# Diverse mating phenotypes impact the spread of *wtf* meiotic drivers in *Schizosaccharomyces pombe*

José Fabricio López Hernández[1], Rachel M Helston[1], Jeffrey J Lange[1], R Blake Billmyre[1], Samantha H Schaffner[1,2], Michael T Eickbush[1], Scott McCroskey[1], Sarah E Zanders[1,3]*

[1]Stowers Institute for Medical Research, Kansas City, United States; [2]Kenyon College, Gambier, United States; [3]Department of Molecular and Integrative Physiology, University of Kansas Medical Center, Kansas City, United States

**Abstract** Meiotic drivers are genetic elements that break Mendel's law of segregation to be transmitted into more than half of the offspring produced by a heterozygote. The success of a driver relies on outcrossing (mating between individuals from distinct lineages) because drivers gain their advantage in heterozygotes. It is, therefore, curious that *Schizosaccharomyces pombe*, a species reported to rarely outcross, harbors many meiotic drivers. To address this paradox, we measured mating phenotypes in *S. pombe* natural isolates. We found that the propensity for cells from distinct clonal lineages to mate varies between natural isolates and can be affected both by cell density and by the available sexual partners. Additionally, we found that the observed levels of preferential mating between cells from the same clonal lineage can slow, but not prevent, the spread of a *wtf* meiotic driver in the absence of additional fitness costs linked to the driver. These analyses reveal parameters critical to understanding the evolution of *S. pombe* and help explain the success of meiotic drivers in this species.

*For correspondence:
sez@stowers.org

## Editor's evaluation

Meiotic drivers are selfish elements that distort segregation to be over-represented in offspring of heterozygotes. Multiple meiotic drive elements are known in the yeast *Schizosaccharomyces pombe*, *S. pombe*, which can seem puzzling as this fungus has long been thought to undergo little outcrossing. This manuscript reports theoretical and experimental analyses suggesting that the outcrossing rate can be high enough in this species to explain the spread of multiple meiotic drive elements. The findings support the emerging view that homothallic fungi can undergo quite high rates of outcrossing, which is also in agreement with evolutionary considerations on the evolution of mating types. This study can thus be of high relevance for scientists studying meiotic drivers and/or mating systems and their evolution.

## Introduction

Mating behaviors have long been a focus of art, literature, and formal scientific inquiry. This interest stems, in part, from the remarkable importance of mate choice on the evolution of species. Outcrossing and inbreeding represent distinct mating strategies that both have potential evolutionary benefits and costs (*Glémin et al., 2019*; *Muller, 1932*; *Otto and Lenormand, 2002*). For example, outcrossing (mating between individuals from distinct lineages) can be beneficial because it can help purge deleterious alleles from a line of descent, but it can also be costly as it can promote the spread of selfish

**eLife digest** The fission yeast, *Schizosaccharomyces pombe*, is a haploid organism, meaning it has a single copy of each of its genes. *S. pombe* cells generally carry one copy of each chromosome and can reproduce clonally by duplicating these chromosomes and then dividing into two cells. However, when the yeast are starving, they can reproduce sexually. This involves two cells mating by fusing together to create a 'diploid zygote', which contains two copies of each gene. The zygote then undergoes 'meiosis', a special type of cell division in which the zygote first duplicates its genome and then divides twice. This results in four haploid spores which are analogous to sperm and eggs in humans that each contain one copy of the genome. The spores will grow and divide normally when conditions improve.

The genes carried by each of the haploid spores depend on the cells that formed the zygote. If the two 'parent' yeast had the same version or 'allele' of a gene, all four spores will have it in their genome. However, if the two parents have different alleles, only 50% of the offspring will carry each version. Although this is usually the case, there are certain alleles, called meiotic drivers, that are transmitted to all offspring even in situations where it is only carried by one parent. Meiotic drivers can be found in many organisms, including mammals, but their behavior is easiest to study in yeast.

Meiotic drivers known as killers achieve this by disposing of any 'sister' spores that do not inherit the same allele of this gene. This 'killing' can only happen when only one of the 'parents' carries the driver. This scenario is thought to rarely occur in species that inbreed, as inbreeding leads to both gene copies being the same. However, this does not appear to be the case for *S. pombe*, which contain a whole family of killer meiotic drivers, the *wtf* genes, despite also being reported to mainly inbreed.

To investigate this contradiction, López Hernández et al. isolated several genetically distinct populations of *S.pombe*. These isolates were grown together to determine how often the each one would outcross (mate with an individual from a different population) or inbreed. The results found that levels of inbreeding varied between isolates.

Next, López Hernández et al. used mathematical modelling and experimental evolution analyses to study how *wtf* drivers spread amongst these populations. This revealed that *wtf* genes spread faster in populations with more outcrossing. In some instances, the *wtf* driver was linked to a gene that could harm the population. In these cases, López Hernández et al. found than inbreeding could purge these drivers and stop them from spreading the dangerous alleles through the population.

López Hernández et al. establish a simple experimental system to model driver evolution and experimentally demonstrate how key parameters, such as outcrossing rates, affect the spread of these genes. Understanding how meiotic drivers spread is important, as these systems could potentially be used to modify populations important to humans, such as crops or disease vectors.

genes (*Burt and Trivers, 1998*; *Crow, 1988*; *Hurst and Werren, 2001*; *McDonald et al., 2016*; *Zeyl et al., 1996*).

Meiotic drivers represent one type of selfish genetic element that relies on outcrossing to persist and spread in a population (*Lindholm et al., 2016*; *Sandler and Novitski, 1957*). These loci can manipulate the process of gametogenesis to bias their own transmission into gametes at the expense of the rest of the genome (*Burt and Trivers, 2006*). Meiotic drivers are often considered selfish or parasitic genes because they generally offer no fitness benefits to their hosts and are instead often deleterious or linked to deleterious alleles (*Dyer et al., 2007*; *Higgins et al., 2018*; *Klein et al., 1984*; *Rick, 1966*; *Schimenti et al., 2005*; *Taylor et al., 1999*; *Wilkinson and Fry, 2001*). As inbreeding is thought to inhibit the spread of selfish genes like drivers, drivers are predicted to be unsuccessful in species that rarely outcross (*Burt and Trivers, 1998*; *Hurst and Werren, 2001*). This assumption appears to be challenged in several fungal species, including the fission yeast *Schizosaccharomyces pombe* (*Grognet et al., 2014*; *Hammond et al., 2012*; *Svedberg et al., 2021*; *van der Gaag et al., 2000*; *Vogan et al., 2021*).

Generally, *S. pombe* cells grow as haploids and their mating (fusion of two haploids of opposite mating type) generates diploid zygotes that can then undergo meiosis to generate haploid progeny known as spores (*Figure 1—figure supplement 1A*). Mating in *S. pombe* is widely thought

to preferentially occur between daughter cells clonally derived from a common progenitor via a recent mitotic division, which we refer to as 'same-clone mating' (*Figure 1—figure supplement 1B*; *Egel, 1977*; *Gutz and Doe, 1975*; *Miyata and Miyata, 1981*; *Billiard et al., 2012*; *Perrin, 2011*). The support for this idea is described below. Despite this notion that *S. pombe* cells preferentially undergo same-clone mating (leading to inbreeding), *S. pombe* isolates host multiple meiotic drivers (*Bravo Núñez et al., 2020b*; *Eickbush et al., 2019*; *Farlow et al., 2015*; *Hu et al., 2017*; *Nuckolls et al., 2017*; *Zanders et al., 2014*).

In the wild, a minority of *S. pombe* strains are heterothallic, meaning they have a fixed mating type (*h+* or *h-*; *Gutz and Doe, 1975*; *Nieuwenhuis and Immler, 2016*; *Schlake and Gutz, 1993*). Heterothallic *S. pombe* strains must mate with non-clonally related cells of the opposite mating type to complete sexual reproduction (*Egel, 1977*; *Gutz and Doe, 1975*; *Leupold, 1949*; *Miyata and Miyata, 1981*; *Nieuwenhuis and Immler, 2016*; *Osterwalder, 1924*; *Schlake and Gutz, 1993*). However, most isolates of *S. pombe* are homothallic, that is, they can switch between the two mating types (*h+* and *h-*) during clonal expansion via mitosis (*Figure 1—figure supplement 1B*; *Egel and Eie, 1987*; *Gutz and Doe, 1975*; *Klar, 1990*; *Nieuwenhuis et al., 2018*; *Singh and Klar, 2003*). Homothallism in *S. pombe* enables mating to occur between cells clonally derived from the same progenitor via mitosis (*Billiard et al., 2012*). Mating can even occur between two cells produced by a single mitotic cell division (*Egel, 1977*; *Gutz and Doe, 1975*; *Miyata and Miyata, 1981*).

It is often assumed that homothallic fungi, like *S. pombe*, do not generally mate with non-clonally related cells. It is important to note, however, that homothallism in fungi does not inherently lead to mating between clonally related cells (*Giraud et al., 2008*). Under some conditions, such as when gametes are dispersed and finding a mate is costly, homothallism can be a beneficial strategy for outcrossing species as it ensures gamete compatibility (*Billiard et al., 2012*). In addition, population genetic analyses of one homothallic fungus, *Sclerotinia sclerotiorum*, support that homothallism is compatible with frequent outcrossing (*Attanayake et al., 2014*).

Very little is known about the ecology of *S. pombe* in the wild, including how often *S. pombe* cells outcross (*Jeffares, 2018*). In the lab, the mating propensities of *S. pombe* have mostly been investigated in derivatives of a strain first isolated from French wine in 1921 (*Osterwalder, 1924*). In this work, we will refer to derivatives of this isolate as '*Sp*'. Distinct homothallic *Sp* strains can be induced to mate in the lab (i.e., outcrossed), but such crosses are significantly more challenging to execute than crosses between heterothallic isolates. The relative difficulty associated with outcrossing homothallic strains has been attributed to preferential same-clone mating (*Ekwall and Thon, 2017*; *Forsburg and Rhind, 2006*). Microscopy experiments have supported the idea that homothallic *Sp* haploids tend to undergo same-clone mating (*Bendezú and Martin, 2013*; *Egel, 1977*; *Leupold, 1949*; *Miyata and Miyata, 1981*). However, the precise level of same-clone mating when homothallic *S. pombe* cells are among nonclonal sexual partners has not, to our knowledge, been formally reported for any isolate.

Population genetic analyses have provided additional support for the notion that *S. pombe* inbreeds. There are two major *S. pombe* lineages that diverged between 2300 and 78,000 years ago (*Tao et al., 2019*; *Tusso et al., 2019*). Much of the sampled variation within the species represents different admixed hybrids of those two ancestral lineages resulting from an estimated 20–60 outcrossing events (*Tusso et al., 2019*). This low outcrossing rate could result from limited opportunities to outcross, preferential same-clone mating, or a combination of the two. It is important to note, however, that the outcrossing rate estimates in *S. pombe* are likely low as pervasive meiotic drive and decreased recombination in heterozygotes suppress the genetic signatures that are used to infer outcrossing (*Bravo Núñez et al., 2020b*; *Hu et al., 2017*; *Nuckolls et al., 2017*; *Zanders et al., 2014*).

Accounting for all these factors to generate a more accurate *S. pombe* outcrossing rate from population genetic data is currently an insurmountable obstacle for several reasons. First, the landscape of meiotic drivers and their suppressors is complex in that more than four drivers and more than eight suppressors of drive could be present in a given heterozygote produced by outcrossing (*Eickbush et al., 2019*; *Bravo Núñez et al., 2020a*). Predicting how those factors will affect allele transmission is well beyond the capabilities of current population genetic models, even if the epistatic relationships within the drive systems were understood, which they currently are not. In addition to the drivers, *S. pombe* isolates contain an array of chromosomal rearrangements and other unknown barriers to recombination (*Avelar et al., 2013*; *Brown et al., 2011*; *Jeffares et al., 2017*; *Tusso et al., 2019*; *Zanders et al., 2014*). Importantly, recombination rates are also inextricably linked to the complex

drive landscape because the presence of drivers also affects the recombination rate within viable offspring, independent of chromosomal rearrangements (*Bravo Núñez et al., 2020b*).

Given those limitations, the current laboratory knowledge and outcrossing rate estimates suggest that drivers would infrequently have the opportunity to act in *S. pombe* (*Egel, 1977*; *Ekwall and Thon, 2017*; *Forsburg and Rhind, 2006*; *Gutz and Doe, 1975*; *Miyata and Miyata, 1981*; *Tusso et al., 2019*). In fact, the notion that *S. pombe* infrequently outcrosses has even been used to challenge the idea that *S. pombe* meiotic drivers are truly selfish genes that persist due to meiotic drive (*Sweigart et al., 2019*). Nonetheless, the *S. pombe* genome houses numerous meiotic drive genes from the *wtf* gene family (*Bravo Núñez et al., 2018*; *Bravo Núñez et al., 2020a*; *Eickbush et al., 2019*; *Hu et al., 2017*; *Nuckolls et al., 2017*).

The *wtf* drivers destroy the meiotic products (spores) that do not inherit the driver from a heterozygote. Each *wtf* drive gene encodes both a Wtf$^{poison}$ and a Wtf$^{antidote}$ protein that, together, execute targeted killing of the spores that do not inherit the *wtf* driver (*Hu et al., 2017*; *Nuckolls et al., 2017*). In the characterized *wtf4* driver, the Wtf4$^{poison}$ protein assembles into toxic protein aggregates that are packaged into all developing spores. The Wtf4$^{antidote}$ protein co-assembles with Wtf4$^{poison}$ only in the spores that inherit *wtf4* and likely neutralizes the poison by promoting its trafficking to the vacuole (*Nuckolls et al., 2020*). Spore killing by *wtf* drivers leads to the loss of about half of the spores and almost exclusive transmission (>90%) of the *wtf* driver from a heterozygote (*Hu et al., 2017*; *Nuckolls et al., 2017*). Despite their heavy costs to the fitness of heterozygotes, the drivers are successful in that all assayed *S. pombe* isolates contain multiple *wtf* drivers (4–14 drivers; *Bravo Núñez et al., 2020a*; *Eickbush et al., 2019*; *Hu et al., 2017*). In addition to those drivers, the *S. pombe* isolates also contain between 8 and 17 suppressors of drive that encode only a Wtf$^{antidote}$ protein.

In this work, we exploited the tractability of *S. pombe* to better understand how mating phenotypes, particularly inbreeding propensity, could affect the spread of a meiotic driver in this species. Despite limited genetic diversity among isolates, we observed natural variation in inbreeding propensity and other mating phenotypes. Some natural isolates preferentially undergo same-clone mating in the presence of a potential outcrossing partner, whereas others mate more randomly. Additionally, we found that the level of same-clone mating can be altered by cell density and affected by the available sexual partners. This is important as it highlights that our measured values in the lab are not meant to indicate precise levels of outcrossing that would occur under unknown natural conditions.

To explore the effects that varying mating phenotypes could have on the spread of a *wtf* driver in a population, we used both mathematical modeling and an experimental evolution approach. We found that, while the spread of a *wtf* driver could be slowed by the observed levels of same-clone mating, the driver could still spread in the absence of linked deleterious traits. We incorporated our observations into a model in which rapid *wtf* gene evolution and occasional outcrossing facilitate the maintenance of *wtf* drivers. More broadly, this study illustrates how the success of drive systems is impacted by mating phenotypes.

## Results

### Same-clone mating propensity differs among *S. pombe* natural isolates

To quantify mating propensities of homothallic *S. pombe* strains in the presence of other potential homothallic sexual partners, we first generated fluorescently tagged strains to easily observe mating via microscopy (*Figure 1A*). We marked strains with either GFP or mCherry (both constitutively expressed and integrated at the *ura4* locus). We then mixed equal proportions of GFP-expressing and mCherry-expressing haploid cells and plated them on a medium (SPA) that induces one round of mating and meiosis. We imaged the cells immediately after plating to measure the starting frequency of both parent types. We then imaged again 24–48 hr later when many cells in the population had mated to either form zygotes or fully developed spores. We inferred the genotypes (homozygous or heterozygous) of each zygote and ascus (spore sac) based on their fluorescence (*Figure 1A*). Homozygotes were produced by mating of two cells carrying the same fluorophore, while heterozygotes were produced by mating between a GFP-labeled and an mCherry-labeled cell (*Figure 1B and C*). Finally, we calculated the inbreeding coefficient (*F*) by comparing the observed frequency of heterozygotes to the frequency expected if mating was random (*F* = 1–observed heterozygotes/expected heterozygotes; *Figure 1A*). Exclusive mating between cells with the same fluorophore, random mating, and

**A**

Mix haploid cells

GFP

mCh

SPA 0 h

Mating+Meiosis

24-48 h

$$F = 1 - \frac{Observed\ heterozygotes}{Expected\ heterozygotes}$$

Microscope

Zygotes   Asci

Homozygotes

Heterozygotes

**B** *S. pombe* (*Sp*, Lab strain)

**C** *S. kambucha* (*Sk*)

**D**

Inbreeding coefficient (*F*)

Complete inbreeding

Random mating

FY29043  *Sp*  FY29022  FY28981  FY28974  FY29044  *Sk*  *Sp* Heterothallic

Homothallic natural isolates

**E** Mating efficiency %

| | |
|---|---|
| FY29043 | 33±2 |
| *Sp* | 33±1 |
| FY29022 | 25±3 |
| FY28981 | 10±4 |
| FY28974 | 23±7 |
| FY29044 | 14±2 |
| *Sk* | 50±2 |

**Figure 1.** Inbreeding coefficients vary between homothallic isolates of *Schizosaccharomyces pombe*.
 (**A**) Experimental strategy to quantify mating patterns between mixed homothallic isolates. GFP (cyan)- and mCherry (magenta)-expressing cells were mixed and placed on SPA medium that induces mating and meiosis. An agar punch from this plate was imaged to assess the initial frequencies of each haploid strain. After incubation at 25°C for at least 24 hr, another punch was imaged to determine the number of homozygous and heterozygous zygotes/asci based on their fluorescence. The inbreeding coefficient (**F**) was calculated using the formula shown. (**B and C**) Representative images of the mating in *Sp* (**B**) and *Sk* (**C**) isolates after 24 hr. Filled arrowheads highlight examples of homozygous asci whereas open arrowheads highlight

*Figure 1 continued on next page*

Figure 1 continued

heterozygous asci. A few additional zygotes are also outlined with dotted lines in the images. Scale bars represent 10 μm. (**D**) Inbreeding coefficient of homothallic natural isolates and complementary heterothallic (*h+,h-* mCherry and *h+,h-* GFP) *Sp* lab strains. At least three biological replicates per isolate are shown (open shapes). (**E**) Mating efficiency of the isolates shown in (**D**) (%) ± standard error from three biological replicates of each natural isolate.

The online version of this article includes the following source data and figure supplement(s) for figure 1:

**Source data 1.** *Schizosaccharomyces pombe* natural isolates.

**Figure supplement 1.** Sexual cycle of *Schizosaccharomyces pombe*.

**Figure supplement 2.** Inbreeding coefficients can be affected by cell density.

**Figure supplement 2—source data 1.** Raw data of allele transmission values reported in *Figure 1—figure supplement 2*.

**Figure supplement 3.** Homothallism of natural isolates.

**Figure supplement 4.** Density variation in plated cells.

exclusive mating between cells with different fluorophores would yield coefficients of 1, 0, and –1, respectively (*Hartl and Clark, 2007*). This assay occurs in a specific laboratory context, so the conditions differ from those in the wild. Importantly, however, we assay all strains under the same conditions, so the assay is sufficient to explore possible variation in mating phenotypes between different isolates of *S. pombe*.

In homothallic cells of the common lab isolate, *Sp*, we measured an average inbreeding coefficient of 0.57 using our microscopy assay (*Figure 1B and D*). As a control, we also assayed a mixed heterothallic population containing roughly equal amounts of GFP-expressing and mCherry-expressing cells of both mating types. We expected this control population to exhibit random mating between GFP- and mCherry-expressing cells, as heterothallic cells cannot undergo same-clone mating. We did observe random mating of this mixed heterothallic population ($F$ = −0.05), which helps validate our assay (*Figure 1D*). To further validate our microscopy results, we also assayed *Sp* cells using an orthogonal approach employing traditional genetic markers. For this analysis, we mixed haploid cells on supplemented SPA medium (SPAS) to induce them to mate and undergo meiosis. We then manually genotyped the progeny and used the fraction of recombinant progeny to calculate inbreeding coefficients (*Figure 1—figure supplement 2A*). The average inbreeding coefficients measured using the genetic assay were very similar to the values we measured using the microscopy assay (0.49 for homothallic *Sp* cells; *Figure 1—figure supplement 2B*). Together, our results confirm and quantify previous observations of non-random mating in homothallic *Sp* cells (*Bendezú and Martin, 2013*; *Egel, 1977*). In addition, we demonstrate that our fluorescence assay provides a powerful tool to quantify mating events.

We next extended our analyses to other *S. pombe* natural isolates collected from the wild. We assayed six additional isolates, FY29043, FY29022, FY28981, FY28974, FY29044, and *Schizosaccharomyces kambucha* (*Sk*), using our fluorescence microscopy assay. We chose these isolates because they each contain different fractions of the two inferred ancestral *S. pombe* lineages (*Tusso et al., 2019*). In addition, the strains we chose are homothallic, sporulated well, were non-clumping, and we were able to transform them with the GFP and mCherry markers described above (*Figure 1—source data 1*, *Figure 1—figure supplement 3*; *Jeffares et al., 2015*). We found that the measured inbreeding coefficients varied significantly between the different natural isolates (*Figure 1D*). The phenotype of FY29043 was similar to *Sp*, but other *S. pombe* isolates, including *Sk*, mated more randomly (*Figure 1D*). We also observed variation in mating efficiency ranging from 10% of cells mating in FY28981 to 50% of cells mating in *Sk* (*Figure 1E*).

Given that *S. pombe* cells are immobile, we thought that cell density could affect their propensity to undergo same-clone mating. To test this, we compared the inbreeding coefficients of both homothallic *Sp* and *Sk* isolates at three different starting cell densities: our standard mating density (1×), high density (10×), and low density (0.1×). Because crowding prevented us from assaying high-density cells using our microscopy approach, we used the genetic assay for each condition. We found that inbreeding coefficients were higher in both *Sp* and *Sk* isolates when cell densities were lower (*Figure 1—figure supplement 2C and D*). This is likely because cells plated at low density tended to be physically distant from potential sexual partners that were not part of the same clonally growing patch of cells (*Figure 1—figure supplement 4*). However, a control population consisting of mixed heterothallic *Sp* cells (*h+* and *h-* cells of two different genotypes mixed in equal proportions) that

cannot mate within a clonal patch of cells showed near random mating between the two genotypes at all cell densities assayed (*Figure 1—figure supplement 2C*). Overall, these experiments demonstrate that inbreeding coefficients vary within *S. pombe* homothallic isolates and can be affected by cell density.

## Additional phenotypes associated with reduced inbreeding coefficients in *Sk*

We next used time-lapse imaging to explore the origins of the different inbreeding coefficients, focusing on the *Sp* and *Sk* isolates. Previous work claimed that *Sk* cells have reduced mating-type switching efficiency, based on levels of the DNA break (double-strand break site [DSB]) that initiates switching (*Singh and Klar, 2002*). A mutation at the *mat-M* imprint site was proposed to be responsible for the reduced level of DSBs (*Singh and Klar, 2003*).

We reasoned that less mating-type switching could lead to less same-clone mating, as a small population of cells clonally derived from the same progenitor via mitosis would be less likely to contain cells of compatible mating types. This would promote cells mating outside their clonal lineage. One can see evidence consistent with this phenomenon in our experiments varying the cell density of homothallic *Sp* cells (*Figure 1—figure supplement 2C-D*). Specifically, we observed more random mating when cells were at higher density, which likely results from more opportunity for cells to mate outside their clonal lineage. We aspired to directly compare mating type switching rates between *Sp* and *Sk* cells, although such direct assays have not, to our knowledge, been done in any strain background. We attempted to develop a direct assay using previously described cytological reporters of mating type. The reporters have been used to assay ratios of *h +* and *h-* cells in a population (*Jakočiūnas et al., 2013*, *Maki et al., 2018*), but we found they were not well-suited for following switching events live in time-lapse imaging in our hands (*Figure 2—figure supplement 1A*).

Because of this, we decided to further explore the possible differences in mating-type switching frequencies indirectly using time-lapse assays, similar to those used to originally determine the patterns of mating-type switching in *Sp* (*Miyata and Miyata, 1981*). As in the classic study of *Miyata and Miyata, 1981*, we used the first mating event as an imperfect proxy for when cells were capable of mating (i.e., they had a partner of the opposite mating type). For these assays, we tracked the fate of individual homothallic founder cells plated on SPA at low density (0.25× to our standard mating density used above) to quantify how many mitotic generations occurred prior to the first mating event. When the first mating event occurred, we recorded the proportion of the cells present that mated (prior to the appearance of cells from next mitotic generation). We also recorded the relationships between the cells that did mate (*Figure 2A*). Specifically, we scored whether the mated cells were the product of a single mitotic division. Historically, these have been called 'sister cells' in the *S. pombe* mating literature, although we use the term 'sibling cells' to be gender neutral (e.g., *Miyata and Miyata, 1981*). We did not consider cells that were born in mitotic generations past the one in which mating first occurred (*Figure 2A*). For example, if cells within a given lineage first mated at generation 3, we scored the relationships between those mated cells and recorded how many unmated cells remained at the end of generation 3 (even if descendants of the unmated cells eventually mated). We then did not consider that lineage any further, so lineages in which cells first mated at generation 2 are not represented in the generation 3 data.

Classic work characterizing mating-type switching patterns in *Sp* found that one in a group of four clonally derived cells will have the opposite mating type relative to the other three cells. A newly switched cell will be compatible to mate with close relatives, most typically its sibling cell (*Figure 2A*; *Miyata and Miyata, 1981*). Under the same switching model, a smaller portion of cells derived from a single division could be of opposite mating types and thus sexually compatible. For example, if one considers switchable cells (e.g., Ps in *Figure 2A*), they must divide only once to generate a pair of mating-competent cells. In *Sp* cells, we observed mating among the clonal descendants of some progenitor cells after a single mitotic division (i.e., at generation 2). By the third generation, we observed mating among the descendants of more than half of the progenitor cells. Almost all the observed mating events were between sibling cells (*Figure 2A and B*). These observations are consistent with published work assaying mating patterns within lineages of clonal *Sp* cells (*Bendezú and Martin, 2013*; *Klar, 1990*; *Miyata and Miyata, 1981*).



**Figure 2.** Variation between mating behaviors in *Sp* and *Sk* cells. (**A**) Schematic showing cell divisions, switching, and mate choice in *Sp*. Mitotic generations are shown on the left. Cells are either *h+* (**P**) or *h-* (**M**) and their status is switchable (**s**) or unswitchable (**u**). (**B**) Division and mating phenotypes of wildtype *Sp* and *Sk* cells plated at low cell density (0.25×) on SPA plates. Single founder cells and their descendants were monitored through time-lapse imaging. Mating patterns were categorized until the end of the generation in which the first mating event occurred. The data presented were pooled from two independent time-lapses in which over 400 founder cells were tracked. (**C**) Division and mating phenotypes of fluorescently labeled *Sp* and *Sk* cells plated at standard (1×) density on SPA plates. Individual cells were monitored until the population reached a typical mating efficiency for the specific cross and the mate choice of their progeny and the mitotic generation of those events was classified. The cells labeled with GFP are indicated with a cyan line around the cell whereas the mCherry-labeled cells are outlined in magenta. Pooled data from two independent experiments are presented, with 286 cells scored from each experiment. In the cross of *Sp* (GFP) and *Sk* (mCherry), we present an additional plot (far right) in which the fate of each founder is separated by isolate. (**D**) Inbreeding coefficient calculated from still images of cells plated at 1× density on SPA. *** indicates p-value < 0.005, multiple t-test, Bonferroni corrected. At least three biological replicates per isolate are shown (open shapes). (**E**) Breakdown of sibling cell mating by fluorophore in the indicated crosses from C. * indicates p-value = 0.04, one-tailed t-test comparing isogenic and mixed *Sp* cells from two videos.

*Figure 2 continued on next page*

*Figure 2 continued*

The online version of this article includes the following video, source data, and figure supplement(s) for figure 2:

**Figure supplement 1.** Mating-type switching dual reporter assay using mating-type-specific promoters in *Sp*.

**Figure supplement 1—source data 1.** Original file of the full raw unedited gel shown in *Figure 2—figure supplement 1B*.

**Figure supplement 1—source data 2.** Original file of the full raw unedited gel shown in in *Figure 2—figure supplement 1B* with relevant band labeled.

**Figure supplement 2.** Mating-type locus variation in *Schizosaccharomyces pombe* isolates.

**Figure supplement 3.** Variation in cell division and asci phenotypes in natural isolates.

**Figure supplement 4.** Mating efficiency and asci length of *Sp/Sk* heterozygotes.

**Figure supplement 5.** Mostly additive phenotypes in crosses between natural isolates.

**Figure 2—video 1.** Homothallic *Sp* mating video.

https://elifesciences.org/articles/70812/figures#fig2video1

**Figure 2—video 2.** Homothallic *Sk* mating video.

https://elifesciences.org/articles/70812/figures#fig2video2

---

*Sk* cells plated at 0.25× density on SPA divided significantly more than *Sp* cells prior to the first mating (Wilcox rank sum test; p < 0.005 *Figure 2B*). *Sk* progenitor cells most frequently started mating at the fourth mitotic generation (*Figure 2B*). This phenotype is consistent with less mating-type switching as more generations would be required on average to produce a cell with the opposite mating type (*Singh and Klar, 2002*). In addition, many mating events were between non-sibling cells. This phenotype can also be explained indirectly by reduced mating-type switching. Specifically, after cells undergo several divisions, they generate non-linear cell clusters in which comparably more non-sibling cells are in close proximity (*Figure 2—figure supplement 1A-B*). This clustering could lead to more non-sibling mating than when cells are mating-competent after fewer divisions and the low number of cells are largely arranged linearly.

To better understand the differences between *Sp* and *Sk*, we compared the sequence of the mating-type locus in the two isolates as many distinct alleles have been identified (*Nieuwenhuis et al., 2018*). Consistent with previous work, we found that the mating-type regions of *Sp* and *Sk* are highly similar (*Singh and Klar, 2002*). However, using a previously published mate-pair sequencing dataset, we discovered an ~5 kb insertion of nested *Tf* transposon sequences in the *Sk* mating-type region (*Eickbush et al., 2019*). We confirmed the presence of the insertion using PCR (*Figure 2—figure supplement 2A-B*). We also found evidence consistent with the same insertion in FY28981, which also mates more randomly than *Sp* (*Figure 2—figure supplement 2B*, *Figure 1D*). We did not, however, formally test if the insertion affects mating phenotypes. Even if it does have an effect, it is insufficient to explain all the mating variation we observed as FY29044 mates randomly, yet it lacks the insertion (*Figure 1D* and *Figure 2—figure supplement 2B*).

To further explore the hypothesis that decreased mating-type switching efficiency in *Sk* could contribute to the mating differences we observed (*Figure 2B*), we carried out time-lapse analyses of cells, at our standard 1× cell mating density. We reasoned that at this density, any given cell is likely to have a cell of opposite mating type nearby, even if mating-type switching is infrequent. We again used mixed populations of GFP- and mCherry-expressing cells to facilitate the scoring of mating patterns (*Figure 2C*, *Figure 2—video 1* supplement 1). We found that the *Sp* cells predominantly mated in the second and third mitotic generations and most mating events were between sibling cells (*Figure 2C*).

The mating behavior of *Sk* cells changed more dramatically between 0.25× density and the higher 1× density. Whereas *Sk* cells tended to first mate in the fourth generation at 0.25× density, at 1× density *Sk* cells, like *Sp*, generally mated in the second and third mitotic generations (*Figure 2C*, *Figure 2—video 2* supplement 2). Additionally, we observed significantly reduced levels of mating between *Sk* sibling cells at 1× density relative to 0.25× (10% and 56%, respectively; *Figure 2E and B*). These phenotypes are consistent with reduced mating-type switching in *Sk*. Specifically, our data suggest that *Sk* cells do not need to undergo more divisions before they are competent to mate. Rather, the additional divisions that occurred at 0.25× density in *Sk* could have been necessary to produce a pair of cells with opposite mating types. At 1× density, additional divisions are not expected to be required as additional non-sibling compatible partners are available.

It is important to note, however, that our experiments combined with the previous work showing fewer switching-initiating DSBs in *Sk* (*Singh and Klar, 2002*), support, but do not conclusively demonstrate that mating-type switching occurs less frequently in *Sk*. In particular, our assays to understand switching frequencies use mating as a proxy for switching, which is not ideal. Therefore, reduced switching in *Sk* represents a promising model that remains to be tested. Still, our results conclusively demonstrate the key point that the mating phenotypes previously measured in *Sp* do not apply to all *S. pombe* isolates. Despite very little genetic diversity, *S. pombe* isolates maintain significant natural variation in key mating phenotypes (*Jeffares et al., 2017*).

## Ascus variation

While assaying inbreeding cytologically, we noticed that the *Sk* natural isolate displayed tremendous diversity in ascus size and shape (*Figure 1C*, *Figure 2—figure supplement 3A*, *Figure 2—video 2*). This was due to high variability in the size of the mating projections, known as shmoos. *Sk* produced long shmoos only in response to cells of the opposite mating type and not as a response to nitrogen starvation alone (*Figure 2—figure supplement 3B*). The long *Sk* shmoos motivated us to quantify asci length across all the natural isolates described above. We found that most isolates generated zygotes or asci that were ~10–15 µm, similar to *Sp*. The majority of *Sk* zygotes and asci also fell within this range, but ~25% of *Sk* zygotes and asci were longer than 15 µm, with some exceeding 30 µm (*Figure 2—figure supplement 3C*). We also assayed zygote/ascus length in an additional natural isolate in which we were unable to quantify inbreeding due to a clumping phenotype (FY29033). This isolate also showed populations of long asci, like *Sk* (*Figure 2—figure supplement 3C*).

Additionally, we occasionally noticed a fused asci phenotype in *Sk* (*Figure 2—figure supplement 3D*). Time-lapse analyses of mating patterns, described above, revealed these fused asci can result from an occasional disconnect between mitotic cycles and the physical separation of cells (*Figure 2—figure supplement 3E*). This phenotype is reminiscent of *adg1*, *adg2*, *adg3*, and *agn1* mutants in *Sp* that have defects in cell fission (*Alonso-Nuñez et al., 2005*; *Gould and Simanis, 1997*; *Sipiczki, 2007*). Although we observed this phenotype in all time-lapse experiments using *Sk* cells, the prevalence of this phenotype varied greatly between experiments. We rarely observed this phenotype in *Sp* cells. We did not analyze time-lapse images of the other natural isolates, where this phenotype is most easily observed, so it is unclear if this septation phenotype occurs in other natural isolates.

## Mating phenotypes are affected by available mating partners

We next assayed if the mating preferences of *S. pombe* isolates *Sp* and *Sk* were invariable, or if they could be affected by the available mating partners due to mating incompatibilities (*Seike et al., 2019b*). To test this idea, we used both still and time-lapse imaging of cells mated at 1× density on SPA. For these experiments, we mixed fluorescently labeled *Sp* and *Sk* cells in equal frequencies.

We observed in experiments employing still images that the overall inbreeding coefficient of the mixed *Sk/Sp* population of cells was intermediate between single-isolate crosses and mixed crosses (*Figure 2D*). In time-lapse experiments, we observed that *Sk* cells maintained low levels of mating between sibling cells in the mixed *Sk/Sp* population (9.2% compared to 9.8% in a homogeneous population; *Figure 2E*). Among the *Sp* cells, mating between sibling cells decreased significantly from 56.8% to 29.7% in the mixed mating environment (*Figure 2E*; t-test, p = 0.04). Together, these results suggest that *Sk* cells can interfere with the ability of *Sp* cells to undergo same-clone mating. Although sibling cell mating preference changed, we did not observe a significant decrease in the mating efficiency of *Sp* cells in a mixed *Sp/Sk* population relative to a pure *Sp* population (*Figure 2—figure supplement 4A*). Instead, the mating efficiency in the mixed *Sp/Sk* population was intermediate of those observed in pure *Sp* and *Sk* populations, indicating these isolates do not affect each other's ability to mate.

We were intrigued by the idea that long shmoos (mating projections) of *Sk* could contribute to its ability to disrupt *Sp* sibling mating. We were unable to address this idea directly. We did, however, find that *Sk/Sp* matings produce significantly longer zygotes/asci than either *Sk/Sk* or *Sp/Sp* matings (*Figure 2—figure supplement 4B*). This was true even when we compared *Sk/Sp* zygote/ascus length to the length of heterozygous *Sk/Sk* or heterozygous *Sp/Sp* zygotes/asci. While this result does not prove that long *Sk* shmoos disrupt *Sp* sibling mating, it does show that long shmoos tend to be used in these outcrossing events.

We next extended our analyses by assaying mating efficiency and inbreeding coefficients in all pairwise combinations of *Sp*, *Sk*, FY29043, and FY29044 using still images of mated cells. After adjusting for mating efficiencies and parental inbreeding coefficients (see Materials and methods), the phenotypes we observed in these crosses were mostly additive, in that they were intermediate to the pure parental strain phenotypes (*Figure 2—figure supplement 5*). The two exceptions were in the crosses between *Sk* and the isolates FY20943 and FY20944. *Sk* formed more *Sk*/*Sk* homozygotes than expected in the two crosses (one-tailed t-test, p = 0.043 and p = 0.038, respectively), suggesting that *Sk* cells may not be fully sexually compatible with FY20943 and FY20944 (*Figure 2—figure supplement 5*). However, the magnitude of the effect was small in both cases and the statistical significance is lost after Bonferroni correction for multiple testing. Overall, our observations indicate that mating phenotypes of a given isolate can be affected by different mating partners. Importantly, however, our results suggest mating incompatibilities are unlikely to have a major role in limiting outcrossing within *S. pombe*.

## Population genetics modeling of the effect of inbreeding on a *wtf* meiotic driver

We next wanted to test how the observed range of inbreeding coefficients would affect the spread of a *wtf* driver in a population. To do this, we first used population genetic modeling. We used the meiotic drive model presented by J Crow, but we also introduced an inbreeding coefficient (*Hartl and Clark, 2007*; *Crow, 1991*) (see Materials and methods for a full description of the model). The model considers a population with two possible alleles at the queried locus and no genetic drift. We assumed a *wtf* driver would exhibit 98% drive (transmission to 98% of spores) in heterozygotes based on measured values for the *Sk wtf4* driver (*Nuckolls et al., 2017*). We assumed that all genotypes have the same fitness during haploid cell growth. For diploid cells induced to produce spores, we assumed homozygotes have a fitness of 1 (e.g., maximal fitness), whereas *wtf* driver heterozygotes have a fitness of 0.51, since meiotic drive destroys nearly half of the spores (*Nuckolls et al., 2017*). The inbreeding coefficient dictates the frequency of heterozygotes and thus the frequency at which the *wtf* driver can act. We varied the inbreeding coefficient (*F*) from 1 (all matings generate homozygotes) to –1 (all matings generate heterozygotes).

We used the model to calculate the predicted change in the frequency of a *wtf* driver after only one sexual generation (*Figure 3A*). We also calculated the spread of a *wtf* driver in a population from a 5% starting frequency (*Figure 3B*) and from lower starting frequencies (*Figure 3—figure supplement 1A*) over generations of sexual reproduction. If we used an inbreeding coefficient of 1, the frequency of the driver does not increase after sexual reproduction or spread in a population over time (*Figure 3A and B*). No change in driver frequency was expected because no heterozygotes are produced under this condition, so no drive can occur. The *wtf* driver has the greatest advantage if the inbreeding coefficient is –1, as all matings generate heterozygotes. Under all other conditions, including the range of inbreeding coefficients we measured experimentally in *S. pombe* natural isolates, some heterozygotes form. The *wtf* driver thus increases in frequency over generations of sexual reproduction, even when the driver starts at very low (anything greater than 0) frequencies (*Figure 3A and B* and *Figure 3—figure supplement 1A*, see Materials and methods) (*Crow, 1991*). This model predicts that *wtf* drivers can spread under the modeled conditions if some non-same-clone mating occurs, even if it is infrequent. This observation is consistent with previous theoretical analyses demonstrating that inbreeding can slow the spread of meiotic drivers (*Martinossi-Allibert et al., 2021*).

We next wanted to consider the spread of a *wtf* driver under conditions in which genetic drift could occur (*Figure 3—figure supplement 1B and C*). To do this, we simulated populations of different sizes (10–100,000 total individuals) that started with one driver. We followed the population for 1000 generations of mating and spore formation. We randomly selected surviving 'haploids' to populate the next generation to keep the population size fixed. We assumed that all genotypes have equal fitness during haploid cell growth. For mathematical simplicity, we assumed complete drive and a corresponding fitness of 0.5 in heterozygous diploids. We again assumed that both types of homozygous diploids had a fitness of 1. We also varied the inbreeding coefficient from 0 (random mating) to 1 (all matings generate homozygotes). Surprisingly, we found that the probability of a driver's success was related to population size, similar to the recent results of *Martinossi-Allibert et al., 2021*. Specifically, the driver was most likely to be maintained in the smaller populations and in populations with more random mating due to



**Figure 3.** Inbreeding is predicted to slow the spread of a *wtf* driver. We assumed a 0.98 transmission bias favoring the *wtf* driver and a fitness of 0.51 in *wtf+/wtf-* heterozygotes. We assumed that homozygote fitness is 1. We simulated the spread of a *wtf* driver for 30 generations. (**A**) Change of *wtf* meiotic driver frequency after a single meiosis assuming varying levels of inbreeding. (**B**) The spread of a *wtf* driver in a population over time assuming varying levels of inbreeding.

The online version of this article includes the following figure supplement(s) for figure 3:

**Figure supplement 1.** Spread of a *wtf* driver under varying initial frequencies and population sizes.

the driver's positive effect on its own allele transmission (***Figure 3—figure supplement 1C***). In larger populations in which the drivers started at a lower initial frequency, the drive allele generally took a long time to increase its frequency compared to the alternate allele, or it was lost due to drift. This was especially true when inbreeding coefficients were high (***Figure 3—figure supplement 1C***).

# Inbreeding and linked deleterious alleles can suppress the spread of *wtf* drivers

We next wanted to test if our predictions reflect the behavior of *wtf* drive alleles in a laboratory population of *Sp* cells over many generations. To do this, we constructed an experimental evolution system employing the GFP and mCherry fluorescent markers described above to measure changes in allele frequencies in a population over time using cytometry. To mark drive alleles, we linked the fluorescent markers with the *Sk wtf4* driver and integrated the whole construct at the *ura4* locus in *Sp* (*Nuckolls et al., 2017*). For non-driving alleles, we used GFP or mCherry integrated at the *ura4* locus without a linked *wtf* gene. We call the non-*wtf* alleles 'empty vector'. We started the experimental evolution populations with a defined ratio of GFP- and mCherry-expressing cells. We then induced a subset of the population to mate and sporulate followed by collection and culturing of the progeny (spores). From these cells, we remeasured GFP and mCherry frequencies using flow cytometry, and we initiated the next round of mating and meiosis (*Figure 4A*).

Because our experiments rely on comparing the frequency of GFP- and mCherry-expressing cells over time, we needed to test the relative fitness of the markers. We found that both fluorescent markers were lost from all our experimental populations over time (*Figure 4—figure supplement 1A-B*). This was likely because insertion of the markers disrupted the *ura4* gene and cells that excised the marker reverted the *ura4* mutation and thereby gained a fitness benefit. We therefore only considered fluorescent cells for our analyses and stopped the experiments when more than 95% cells lacked a fluorescent marker. In addition, in one set of experiments we also sorted cells at defined timepoints to remove non-fluorescent cells from our populations (*Figure 4D–E*, *Figure 4—figure supplement 1B* and *Figure 4—figure supplement 2C-D*, described below).

To assay for potential differences in the fitness costs of GFP and mCherry markers, we carried out our analyses in two control populations without drive. One control population lacked the *Sk wtf4* driver while the other had *Sk wtf4* linked to both fluorescent markers. For both types of controls, we analyzed homothallic (inbreeding coefficient ~0.5) and mixed heterothallic (inbreeding coefficient ~0) cell populations (*Figure 1D*). We found in most cases that the number of mCherry-expressing cells increased at the expense of GFP-expressing cells over time (*Figure 4—figure supplement 2A-D*). The notable exception was in heterothallic populations containing *Sk wtf4* linked to both fluorophore alleles, where we did not observe a different cost of the GFP allele compared to mCherry (*Figure 4—figure supplement 2D*). The origin of the differential fitness between mCherry and GFP alleles and why this cost was not observed in the one heterothallic population are both unclear. We did not determine the cause of the differences between them.

To allow us to predict expected allele frequencies more accurately, we wanted to obtain a gross estimate of the fitness cost of the GFP-marked alleles relative to the mCherry-marked alleles in our experiments. To do this, we used the first six generations of data from our control crosses (*Figure 4—figure supplement 2*) to fit a maximum likelihood model in which all parameters were fixed except the fitness values of GFP/mCherry heterozygotes and GFP/GFP homozygotes. We found that the fitness cost of the GFP in homozygotes was 0.234 (see Materials and methods). We found the dominance of the GFP cost was 0.083 (low fitness cost of GFP in GFP/mCherry heterozygotes). We then used these costs, calculated from our controls without meiotic drive, to calculate expected values in our experimental analyses in which drive can occur (*Figure 4B–E*).

For our experiments competing *Sk wtf4* with an empty vector allele, we first assayed populations in which the alleles both started at 50% frequency. In homothallic (inbreeding coefficient ~0.5) populations, we observed that *wtf4* alleles spread in the population over several generations of sexual reproduction. The driver spread faster when linked to mCherry than when linked to GFP, presumably due to the aforementioned fitness costs linked to GFP (*Figure 4B–C*). In both cases, the rate of spread of the allele was very close to our model's predictions if we assumed an inbreeding coefficient of 0.5 (black lines, *Figure 4B–C*) and differed considerably from the model's predictions assuming random mating (inbreeding coefficient = 0; gray lines, *Figure 4B–C*).

We saw similar spread of *wtf4* in homothallic populations in a set of repeat experiments in which we sorted the cell populations twice to remove non-fluorescent cells (*Figure 4D and E*). In these experiments, we also assayed mixed heterothallic populations with an equal mix of GFP- and mCherry-marked cells from both mating types. As described above, these mixed heterothallic populations show random mating (inbreeding coefficient ~0) between GFP- and mCherry-labeled cells (*Figure 1D*). In



**Figure 4.** Inbreeding slows the spread of the *wtf4* meiotic driver in homothallic strains. (**A**) Experimental strategy to monitor allele frequency through multiple generations of sexual reproduction. GFP (green) and mCherry (magenta) markers are used to follow empty vector (EV) alleles or the *wtf4* meiotic driver. Starting allele frequencies and allele frequencies after each round of sexual reproduction were monitored using cytometry. (**B**) Homothallic population with mCherry marking *wtf4* and GFP marking an EV allele. Allele dynamics were predicted using drive and fitness parameters described in the text assuming an inbreeding coefficient of 0.5 (black lines) and random mating (inbreeding coefficient of 0; gray lines). The individual

*Figure 4 continued on next page*

*Figure 4 continued*

spots represent biological replicates and the shaded areas around the lines represent 95% confidence intervals. (**C**) Same experimental setup as in (**B**), but with mCherry marking the EV allele and GFP marking *wtf4*. (**D–E**) Repeats of the experiments shown in (**B**) and (**C**) with two alterations. First, these experiments tracked heterothallic populations (dotted lines) in addition to homothallic cells (solid lines). Second, the populations were sorted by cytometry at generations 3 and 5 (vertical long-dashed lines) to remove non-fluorescent cells.

The online version of this article includes the following figure supplement(s) for figure 4:

**Figure supplement 1.** Loss of fluorescent markers in experimental evolution analyses.

**Figure supplement 2.** Allele transmission in homozygous control homothallic and heterothallic populations.

the mixed heterothallic populations, the *wtf4* driver spread significantly faster than in homothallic cells. In generations 1–6, the spread of *wtf4* was very similar to that predicted by our model if we assumed random mating (inbreeding coefficient = 0). In later generations, our observations did not fit the model well. We suspect extensive loss of fluorescent cells, especially those with mCherry, and the resulting decrease in population size could contribute to this effect (***Figure 4D and E***; ***Figure 4— figure supplement 1***).

We compared our data to a model in which driver heterozygotes have a fitness of 0.51, but we also considered variants of the model in which heterozygotes have fitness greater than 0.5, as can occur with spore killers in *Podospora anserina* (***Vogan et al., 2021***; ***Martinossi-Allibert et al., 2021***). In *S. pombe*, an increase in driver heterozygote fitness could occur if the driving alleles benefit from drive beyond the benefits gained directly by killing spores bearing the alternate allele. For example, the surviving meiotic products could theoretically gain fitness during spore development by scavenging increased resources from the killed meiotic products (***Nauta and Hoekstra, 1993***). We found that increasing heterozygote fitness beyond 0.51 decreased the fit of our data to the model (***Figure 5— figure supplement 2***), suggesting *wtf* drivers do not gain additional benefits beyond killing spores bearing the alternate allele.

Overall, our results demonstrate that our population genetics model with driver heterozygote fitness at 0.51 is good at describing the spread of *wtf4* in our experimental population, particularly in the first few generations. Our results also confirm that inbreeding coefficients near 0.5 slow, but do not stop, the spread of drivers in a population.

Meiotic drivers tend to accumulate linked deleterious alleles in nature (***Atlan et al., 2004***; ***Dyer et al., 2007***; ***Finnegan et al., 2019***; ***Fishman and Saunders, 2008***; ***Higgins et al., 2018***; ***Lyon, 2003***; ***Olds-Clarke, 1997***; ***Schimenti et al., 2005***; ***Unckless et al., 2015***; ***Wilkinson and Fry, 2001***; ***Wu, 1983***). While the linkage of individual drivers to deleterious alleles in *S. pombe* has not been extensively investigated, driving alleles in the *Sk* isolate are linked to a chromosomal translocation that decreases fitness in heterozygotes (***Zanders et al., 2014***). Similar chromosomal rearrangements involving chromosome 3, which houses most *wtf* genes, are common in *S. pombe*, so it is likely drivers are frequently linked to rearrangements (***Bowen et al., 2003***; ***Brown et al., 2011***; ***Avelar et al., 2013***). We therefore used the population genetic model to calculate the ability of a driver to spread when tightly linked to alleles with fitness costs ranging from 0 to 0.8. We also varied the inbreeding coefficient from 0 (random mating) to 1 (no heterozygotes are produced).

We found that, in the absence of additional linked costs, *wtf* drivers are predicted to spread in a population at all initial frequencies greater than 0 (***Figure 5A***). As described above, this spread is slowed by inbreeding, but is not stopped until inbreeding coefficients reach 1 (no heterozygotes are produced). When the driver is burdened by additional fitness costs, it can still spread in a population. Importantly, however, as the fitness costs of linked deleterious alleles increase, the driver must start at a higher initial frequency to spread. If the fitness costs are recessive or close to recessive (dominance coefficient *h* near 0), a driver can invade at low frequencies in a randomly mating population, but a driver would require a higher initial frequency to spread in an inbreeding population (inbreeding coefficients > 0; ***Figure 5A***, ***Figure 5—figure supplement 1A***). If the fitness costs are partially dominant (50% dominance), the drivers require a higher initial frequency to spread, even with random mating (***Figure 5—figure supplement 1B***).

In finite populations in which drift can occur, we also observed that linked costs could limit the maintenance of a driver particularly in populations that inbreed (inbreeding coefficients > 0; ***Figure 5— figure supplement 1C,D***). This is in line with previous modeling and can be explained because the

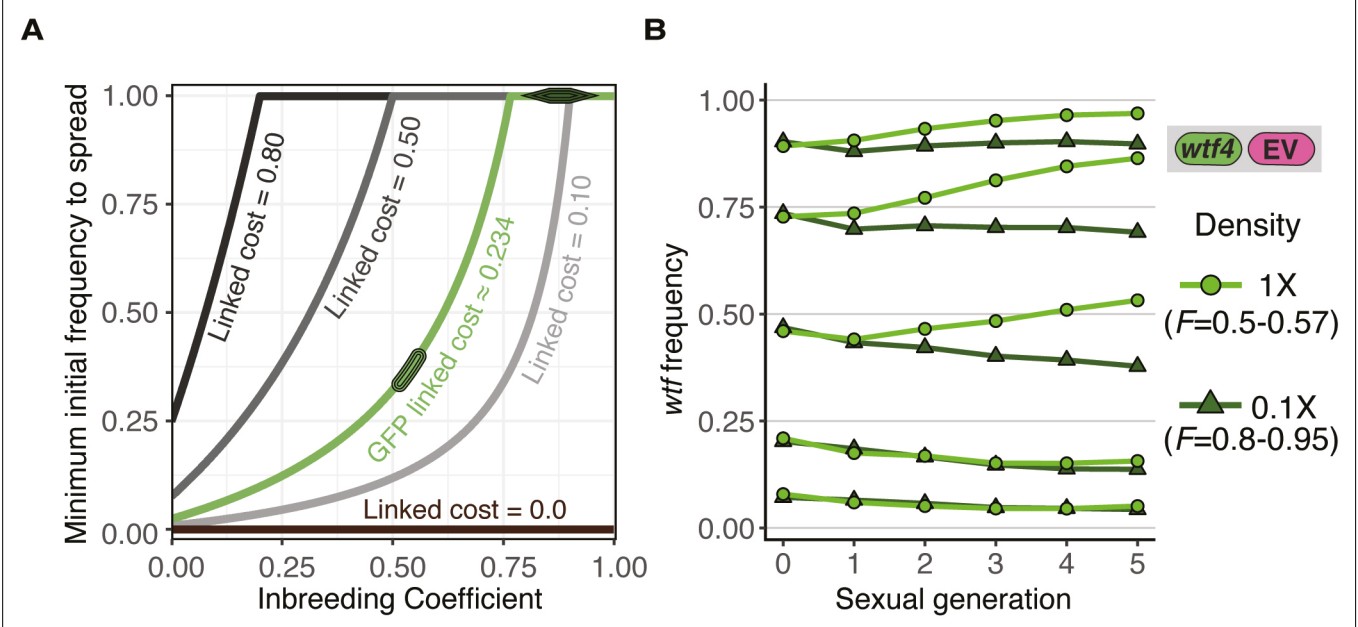

**Figure 5.** Inbreeding purges wtf drivers with linked deleterious alleles. (**A**) Modeling the predicted impacts of inbreeding, the fitness of the driving haplotype, and the initial frequency of the driver on the spread of a *wtf* driver linked to a deleterious allele with low dominance ($h = 0.083$) in a population. The circle indicates the predicted initial frequency necessary for the *GFP:wtf4+* allele to spread in a population when mated at standard (1×) density where $F = 0.5$. The diamond indicates the predicted initial frequency necessary for the *GFP:wtf4+* allele to spread in a population when mated at low (0.1×) density where $F = 0.8–0.95$ measured in *Figure 1—figure supplement 2C*. (**B**) Experimental analyses of the impacts of inbreeding and the initial driver frequency on the spread of the *GFP:wtf4+* allele in a population. A single biological replicate was used for each initial frequency and inbreeding condition.

The online version of this article includes the following figure supplement(s) for figure 5:

**Figure supplement 1.** Evolution of *wtf* meiotic drivers in populations for recessive and codominant costs.

**Figure supplement 2.** Assessing if *wtf+* spores gain fitness as a result of spore killing.

cost of the deleterious allele is fixed, but the benefit a driver gains from drive is frequency dependent (*Drury et al., 2017*; *Martinossi-Allibert et al., 2021*; *Nauta and Hoekstra, 1993*).

We next tested these predictions experimentally using the *Sk wtf4* allele linked to GFP in homothallic cells. As described above, the GFP allele is linked to an unknown deleterious trait (estimated 1.9% and 23.4% cost in heterozygotes and homozygotes, respectively). We varied the inbreeding coefficients of the populations by assaying cells mated at 1× and 0.1× density. As reported above, homothallic cells mated at 1× density exhibit an inbreeding coefficient of 0.5–0.57, but that is increased to 0.8–0.95 by mating the cells at low (0.1×) density (*Figure 1D*, *Figure 1—figure supplement 2B and C*). Consistent with the predictions of our model, we observed in the experimental populations that the driver failed to spread when the initial frequency was less than 0.25 (*Figure 5B*). When the *wtf4:GFP* allele was found in roughly half of the population, it could spread when the inbreeding coefficient was low but decreased in frequency when the inbreeding coefficient was increased (*Figure 5B*). Similar, but less dramatic, effects were observed at higher initial frequencies of the *wtf4:GFP* allele. Altogether, our experimental analyses are consistent with the predictions of our model and show that both inbreeding and linked deleterious alleles can impede the spread of a *wtf* meiotic driver.

## Discussion

### Natural variation in mating phenotypes in *S. pombe*

Mating phenotypes, particularly the outcrossing rate, are key parameters that affect the evolution of species (*Muller, 1932*; *Otto and Lenormand, 2002*). We sought to explore mating phenotypes in *S. pombe* to better understand the evolution of the *wtf* gene family found in this species. Although genetic variation within *S. pombe* is limited, past studies found variation in mating efficiency and

uncovered genetic diversity of the mating-type locus (*Jeffares et al., 2015*; *Nieuwenhuis et al., 2018*; *Rhind et al., 2011*; *Singh and Klar, 2002*). In this work, we assayed mating in an array of homothallic *S. pombe* natural isolates under a variety of laboratory conditions. Similar to previous work, we found *Sp* mating efficiency close to 40% and observed variable mating efficiencies for natural isolates (*Merlini et al., 2016*; *Seike et al., 2019a*). In addition, we quantified the propensity of natural isolates to undergo same-clone mating when given the opportunity to mate with non-clonally related cells. We found this trait, as measured using an inbreeding coefficient, was variable between natural isolates and could be affected by cell density or available sexual partners.

All of the *S. pombe* natural isolates we studied are homothallic and thus capable of same-clone mating. Despite this, we found that all isolates underwent some non-same-clone mating. In addition, some isolates, like *Sk*, showed considerable non-same-clone mating (inbreeding coefficient near 0) under standard mating conditions. This provides additional data supporting homothallism is compatible with outcrossing between non-clonally related isolates (*Attanayake et al., 2014*). In addition, *S. pombe* asci undergo a programmed degeneration process (endolysis) shortly after spore formation (*Encinar del Dedo et al., 2009*). This presumably frees spores to potentially distribute (in wind, water, or associated with animals) separate from the other spores produced in the same meiosis. The physical independence of *S. pombe* spores could also facilitate outcrossing in this homothallic fungus (*Billiard et al., 2012*).

We did not definitively identify the molecular mechanisms underlying the variation in mating phenotypes we observed. Our data is, however, consistent with a model in which less frequent mating-type switching in the *Sk* isolate contributes to more random mating in *Sk* than in the common lab isolate, *Sp*. Specifically, if switching is less frequent in *Sk*, sibling cells are less likely to be compatible to mate. Incompatibility of sibling cells opens the possibility for mating with other, perhaps non-clonally derived, cells in the population. Singh and Klar were the first to propose that *Sk* switched less frequently than *Sp* when they noticed less of the DNA break that initiates switching (*Singh and Klar, 2002*). We discovered a large, nested insertion of transposon sequences in the mating-type locus of *Sk,* and we posit that this insertion could contribute to reduced DNA break formation and, potentially, decreased mating-type switching. It is important to note, however, that not all strains that mate randomly share this transposon insertion. We also stress that, as in previous work (e.g., *Miyata and Miyata, 1981*), we did not directly assay switching rates and instead used mating events to infer information about switching. The long shmoos we observed in *Sk* may also contribute to more random mating in this isolate, as the long shmoo may increase the available number of partners within range.

Additional previously described natural variation that we did not functionally explore may also contribute to differences in inbreeding propensity in *S. pombe*. For example, heterothallic natural isolates are predicted to exclusively undergo non-same-clone mating with cells of the opposite mating type (*Jeffares et al., 2015*; *Nieuwenhuis et al., 2018*). In addition, homothallic isolates with atypical mating-type loci with extra copies of the *mat* cassettes could grow into populations that are biased toward one mating type (*Nieuwenhuis and Immler, 2016*). Indeed, we analyzed the presumably expressed mating-type locus (*mat1*) in several isolates for which we had nanopore sequencing data and found an approximate 3:1 excess of the *h+* allele in FY29033 (*Figure 2—figure supplement 2C*). The excess of one mating type is predicted to also facilitate non-same-clone mating.

It is important to note that our study does not address the actual frequency of outcrossing in *S. pombe* populations in the wild. Very little is known about the ecology of fission yeast, including how frequently genetically distinct isolates are found in close enough proximity to mate (e.g., closer than ~40 μm apart) (*Jeffares, 2018*). Outcrossing rates have been estimated using genomic data, but those estimates generally assume both that heterozygous recombination rates will match those observed in pure *Sp* and that allele transmission is Mendelian (*Farlow et al., 2015*; *Tusso et al., 2019*). Although these genomic estimates are reasonable, neither of these assumptions is consistent with empirical analyses (*Bravo Núñez et al., 2020b*; *Hu et al., 2017*; *Zanders et al., 2014*). These assumptions have, therefore, likely led to an underestimation of the true outcrossing rate.

## The effect of mating-type phenotypes on the spread of *wtf* meiotic drivers

To understand the evolution of the *wtf* drive genes, it is not necessarily essential to understand how frequently significantly diverged natural isolates, like those assayed in this work, mate. Instead, it

is important to understand how often a driver is found in a heterozygous state. We suspect that heterozygosity for *wtf* drivers does not absolutely require outcrossing between more distantly related strains. This is because the *wtf* gene family, particularly the genes involved in meiotic drive, exhibit extremely rapid evolution. Even though genetic diversity within *S. pombe* is low (<1% average DNA sequence divergence in non-repetitive regions), the *wtf* genes present in different isolates tend to be largely distinct (*Eickbush et al., 2019*; *Hu et al., 2017*; *Jeffares et al., 2015*; *Rhind et al., 2011*). The number of *wtf* genes per isolate varies from 25 to 38 *wtf* genes (including pseudogenes), and even genes found at the same locus can be dramatically different (e.g., <61% coding sequence identity between alleles of *wtf24*) (*Eickbush et al., 2019*). For example, *wtf22* is a predicted pseudogene in one strain, a predicted antidote in another strain, and is predicted to encode distinct drivers (i.e., mutually killing) in two more strains. There is only one *wtf* locus where all four strains that were surveyed each contain a meiotic driver, *wtf4* (*Eickbush et al., 2019*). Still, we do not consider this a fixed driver because the sequence of *wtf4* is different in each strain, which, in all cases tested, leads to a distinct drive phenotype (*Bravo Núñez et al., 2018*; *Bravo Núñez et al., 2020a*; *Bravo Núñez et al., 2020b*; *Eickbush et al., 2019*; *Hu et al., 2017*). For example, because *Sk wtf4* and *Sp wtf4* have different sequences, the antidote of *Sk wtf4* does not neutralize *Sp wtf4* and vice versa (*Bravo Núñez et al., 2020a*). The rapid evolution of *wtf* genes is driven largely by non-allelic gene conversion within the family and expansion or contraction of repetitive sequences within the coding sequences of the genes (*Eickbush et al., 2019*).

Because *wtf* genes generally provide no protection against *wtf* drivers with distinct sequences, the variation in *wtf* gene sequences has profound consequences (*Bravo Núñez et al., 2018*; *Bravo Núñez et al., 2020a*; *Hu et al., 2017*). Even small sequence changes in *wtf* drivers can cause the birth and death of drivers. When a cell bearing a novel *wtf* driver mutation mates with a cell without the mutation, the driver is heterozygous, and thus, drive can occur and the novel allele can potentially spread through the otherwise largely homogeneous population. Given fission yeast cells have no inherent mobility, the novel drive alleles could arise within small isolated subpopulations (demes) in which drivers have an increased possibility of establishing, as was formally described by *Martinossi-Allibert et al., 2021*.

Previous work assayed the strength of drive and the associated fitness reduction due to heterozygous *wtf* drivers (*Bravo Núñez et al., 2018*; *Bravo Núñez et al., 2020a*; *Hu et al., 2017*; *Nuckolls et al., 2017*). Those data, along with the inbreeding coefficients measured in this study, allowed us to mathematically model the spread of a *wtf* meiotic driver in an *S. pombe* population. Our modeling showed that the inbreeding coefficients we observed in *S. pombe* could slow the spread of a *wtf* driver. In the absence of drift, however, even the highest inbreeding coefficients we observed in *S. pombe* do not halt the spread of a driver, except in cases where the driver is found in low frequencies and linked to a deleterious allele. Given the tractability of *S. pombe*, we were also able to test the predictions of the model experimentally. Overall, our experimental results were quite similar to the model's predictions discussed above. This suggests that our model encompasses all critical parameters. In addition, our experiments show how the *wtf* drivers can persist and spread in *S. pombe*, even if outcrossing is infrequent. The variation of mating phenotypes also indicates that the rate of spread of a *wtf* driver is expected to vary between different populations of *S. pombe*.

Overall, our results are consistent with previous empirical and modeling studies of meiotic driver dynamics in populations. For example, like our fortuitously deleterious GFP allele, meiotic drivers are often linked to deleterious alleles that can hitchhike with the driver (*Atlan et al., 2004*; *Dyer et al., 2007*; *Finnegan et al., 2019*; *Fishman and Saunders, 2008*; *Higgins et al., 2018*; *Lyon, 2003*; *Olds-Clarke, 1997*; *Schimenti et al., 2005*; *Unckless et al., 2015*; *Wilkinson and Fry, 2001*; *Wu, 1983*). The added costs reduce the spread of drivers, which can lead a population to harbor a driver at stable intermediate frequency (*Dyer and Hall, 2019*; *Finnegan et al., 2019*; *Fishman and Kelly, 2015*; *Hall and Dawe, 2018*; *Manser et al., 2011*). Additionally, inbreeding can be selected as it increases fitness in a population when a driver is recessive lethal (cost ~1, $h = 0$), such as in synthetic drive systems (*Bull, 2016*).

## *S. pombe* as a tool to experimentally model complex drive dynamics

To conclude, we would like to highlight the potential usefulness of the *S. pombe* experimental evolution approach developed for this study. With this system, we were able to observe the effects of

altering allele frequencies, inbreeding rate, and fitness of a driving haplotype. In the future, this system could be used to experimentally explore additional questions about drive systems. For example, one could experimentally model meiotic drivers that bias sex ratios by linking the driver to the mating-type locus in a heterothallic population. In addition, one could explore the evolution of complex multi-locus drive systems employing combinations of multiple *wtf* meiotic drivers or drivers and suppressors. This tool could lead to novel insights about natural drivers, but it may also be particularly useful for exploring potential evolutionary trajectories of artificial gene drive systems (*Burt and Crisanti, 2018*; *Drury et al., 2017*; *Price et al., 2020*; *Wedell et al., 2019*).

## Materials and methods

### Generation of *Ura4*-integrating vectors and fluorescent strains

We introduced the fluorescent genetic markers into the genome using plasmids that integrated at the *ura4* locus. To generate the integrating plasmids, we first ordered gBlocks from IDT (Coralville, IA) that contained mCherry or GFP under the control of a *TEF* promoter and *ADH1* terminator (*Hailey et al., 2002*; *Sheff and Thorn, 2004*). We digested the gBlocks with SpeI and ligated the GFP cassette into the SpeI site of pSZB331 and the mCherry cassette into the SpeI site of pSEZB332 (alternate clone of pSZB331; *Bravo Núñez et al., 2020a*; *Bravo Núñez et al., 2020b*) to generate pSZB437 and pSZB882, respectively. We then linearized the plasmids with KpnI and transformed them into *S. pombe* using the standard lithium acetate protocol (*Schiestl and Gietz, 1989*). We independently transformed the isolates GP50 (*S. pombe*), *S. kambucha*, FY28974, FY28981, FY29022, FY29033, FY29043, and FY29044. We were unsuccessful in transforming FY28969, FY29048, and FY29068. FY29033 was not included in the inbreeding analyses due to its proclivity to clump. The homothallic and heterothallic strains carrying mCherry or GFP were transformed using the same method.

To add *Sk wtf4* to the *Sp* genome, we again used a *ura4*-integrating plasmid. To generate this plasmid, we amplified *Sk wtf4* from SZY13 using the oligos 688 and 686. We digested the amplicon with SacI and ligated into the SacI site of pSZB332 to generate pSZB716 (*Bravo Núñez et al., 2020a*; *Bravo Núñez et al., 2020b*). We then separately introduced the GFP and mCherry gBlocks into the SpeI site of pSZB716 to generate pSZB904 and pSZB909, respectively. We introduced the resulting plasmids into yeast as described above.

### Crosses

We performed crosses using standard approaches (*Smith, 2009*). We cultured each haploid parent to saturation in 3 mL YEL (0.5% yeast extract, 3% dextrose, and 250 mg/L adenine, histidine, leucine, lysine, and uracil) for 24 hr at 32°C. We then mixed an equal volume of each parent (700 µL each for individual homothallic strain, 350 µL for heterothallic parents), pelleted and resuspended in an equal volume of ddH$_2$O (1.4 mL total), then plated 200 µL on SPA (1% glucose, 7.3 mM KH$_2$PO$_4$, vitamins, and agar) for microscopy experiments or SPAS (SPA +45 mg/L adenine, histidine, leucine, lysine, and uracil) for genetics experiments. We incubated the plates at 25°C for 1–4 days, depending on the experiment (see figure legends for exact timing). When we genotyped spore progeny, we scraped cells off of the plates and isolated spores after treatment with B-Glucuronidase (Sigma) and ethanol as described in *Smith, 2009*.

### Iodine staining

We grew haploid isolates to saturation in 3 mL YEL overnight at 32°C. We washed the cells once with ddH$_2$O then resuspended them in an equal volume ddH$_2$O. We then spotted 10 µL of each strain onto an SPAS plate, which we then incubated at 25°C for 4 days prior to staining with iodine (VWR iodine crystals) vapor (*Forsburg and Rhind, 2006*).

### Mating-type locus assembly and PCR

We used mate-pair Illumina sequencing reads to assemble the mating-type locus of *S. kambucha* with previously published data (*Eickbush et al., 2019*). We assembled the mating-type locus using Geneious Prime software (https://www.geneious.com; last accessed March 18, 2019) using an analogous approach to that described to assemble *wtf* loci (*Eickbush et al., 2019*).

## DNA extraction for nanopore sequencing

To extract DNA for nanopore sequencing, we used a modified version of a previously developed protocol (*Jain et al., 2018*). We pelleted 50 mL of a saturated culture and proceeded as described, with the addition of 0.5 mg/mL zymolyase to the TLB buffer immediately prior to use.

## Nanopore sequencing and assembly

We used a MinION instrument and R9 MinION flow cells for sequencing. For library preparation, we used the standard ligation sequencing prep (SQK-LSK109), including end repair using the NEB End Prep Enzyme, FFPE prep using the NEB FFPE DNA repair mix, and ligation using NEB Quick Ligase. We did not barcode samples and thus used each flow cell for a single genome. We used guppy v2.1.3 for base calling. We removed sequencing adapters from the reads using porechop v0.2.2 and then filtered the reads using filtlong v0.2.0 to keep the 100× longest reads. We then error corrected those reads, trimmed the reads and de novo assembled them using canu v1.8 and the ovl overlapper with a predicted genome size of 13 mb and a corrected error rate of 0.12 (*Koren S et al., 2017*). Base called reads are available as fastq files at the SRA under project accession number PRJNA732453.

In order to count allele frequency within the active *mat* locus, we mapped raw reads back to the corresponding de novo assembly using graphmap v0.5.2 and processed using samtools v1.12 (*Li et al., 2009*; *Sović et al., 2016*). We then visually observed the reference-based assemblies using IGV v2.3.97 to count the number of *h* + and *h*- alleles present at the active mating type locus with anchors to unique sequence outside the *mat* locus (*Robinson et al., 2011*).

## Measuring inbreeding coefficients by microscopy

We mixed haploid parents (a GFP- and an mCherry-expressing strain) in equal proportions on SPA as described above. We then left the plate to dry for 30 min and then took a punch of agar from the plate using a 1271E Arch Punch (General Tools, Amazon). We then inverted the punch of agar into a 35 mm glass bottomed dish (No 1.5 MatTek Corporation). We used this sample to count the initial frequency of the two parental types. We then imaged a second punch of agar taken from the same SPA plate after 24 hr incubation at 25°C for homothallic cells and 48 hr for heterothallic cells. Each biological replicate was constituted by a separate cross.

To image the cells, we used an AXIO Observer.Z1 (Zeiss) widefield microscope with a 40× C-Apochromat (1.2 NA) water-immersion objective. We excited mCherry using a 530–585 nm bandpass filter which was reflected off an FT 600 dichroic filter into the objective and collected emission using a longpass 615 nm filter. To excite GFP, we used a 440–470 nm bandpass filter, reflected the beam off an FT 495 nm dichroic filter into the objective and collected emission using a 525–550 nm bandpass filter. We collected emission onto a Hamamatsu ORCA Flash 4.0 using μManager software. We imaged at least three different fields for each sample as each technical replicate.

We used cell shape to identify mated cells (zygotes and asci) and used fluorescence to identify the genotype of each haploid parent. To measure both fluorescence and the length of asci, we used Fiji (https://imagej.net/Fiji) software to hand-draw five pixel-width lines through the length of each zygote or ascus. After subtracting background using a rolling ball background subtraction with width 50 pixels, we then measured the average intensity for the GFP and mCherry channels. When measuring the log10 ratio of GFP over mCherry, the mCherry homozygotes have the lowest ratio, homozygotes for GFP the highest ratio, and heterozygotes intermediate.

To calculate the inbreeding coefficient, we used the formula $F = 1 - $ (observed heterozygotes/ expected heterozygotes). We used Hardy-Weinberg expectations to calculate the expected frequency of heterozygotes ($2p(1p)$) for each sample, where '$p$' is the fraction of mCherry+ cells and $(1-p)$ is the fraction of GFP+ cells measured prior to mating (*Hartl and Clark, 2007*).

## Visualizing mating and meiosis using time-lapse microscopy

For time-lapse imaging of cells mated at 1× density (*Figure 2C*), we prepared cells using the agar punch method described above. For cells at 0.25× density (*Figure 2B*), we used the same approach, except we cultured cells in 3 mL EMM (14.7 mM $C_8H_5KO_4$, 15.5 mM $Na_2HPO_4$, 93.5 mM $NH_4Cl$, 2% w/v glucose, salts, vitamins, minerals) then washed three times with PM-N (8 mM $NA_2HPO_4$, 1% glucose, EMM2 salts, vitamins, and minerals) before plating cells to SPA. While imaging the cells, we added a moistened kimwipe to the MatTek dish to maintain humidity. We sealed the dish lids on with

high-vacuum grease (Corning). We imaged cells using either a Ti Eclipse (Nikon) coupled to a CSU W1 Spinning Disk (Yokagawa), or a Ti2 (Nikon) widefield using the 60× oil immersion objective (NA 1.45), acquiring images every ten minutes for 24–48 hr, using a 5 × 5 grid pattern with 10% overlap between fields. The Ti Eclipse was used for one replicate each of the 1× crosses and the Ti2 was used for all remaining experiments. We used an Okolab stage top incubator to maintain the temperature at 25°C. For the Ti2 (widefield) data we excited GFP through a 470/24 nm excitation filter and collected through an ET515/30 m emission filter. For mCherry on this system, we excited through a 550/15 nm excitation filter and collected through an ET595/40 m emission filter. For the Ti Eclipse (confocal) data, we excited GFP with a 488 nm laser and collected its emission through an ET525/36 m emission filter. For mCherry on this system, we excited with a 561 nm laser and collected through an ET605/70 m emission filter.

To monitor mating in 1× crosses (*Figure 2C-E*), we recorded the number of divisions and mating choice of the progeny of 286 cells until an expected mating efficiency for the population being filmed was attained. The expected mating efficiency was calculated from still images of the same crosses. We recorded two videos of each cross.

To monitor the number of divisions required before mating could occur in 0.25× cultures (*Figure 2B*), around 200 individual cells were monitored through the duration of the generation in which the first mating event occurred. If cells failed to mate, they were monitored throughout the duration of the movie. If a cell or its mitotic offspring interacted with a neighboring cell cluster, it was not included in the analysis. We recorded two videos for each isolate.

## Calculation of mating efficiency

We calculated mating efficiency from microscopic images using the following formula:

$$Mating\ Efficiency\ (\%) = \frac{2Z + 2A + \frac{S}{2}}{V + 2Z + 2A + \frac{S}{2}} * 100$$

where $Z$ represents the number of zygotes, $A$ represents the number of asci, $S$ represents the number of free spores, and $V$ represents the number of vegetative cells (*Seike and Niki, 2017*).

## Switching assay

In an attempt to directly observe mating-type switching, we used the strain TP220 (*Jakočiūnas et al., 2013*). This strain contains a dual reporter system with YFP under control of the *h-* cell-specific *mfm3* promoter and CFP under control of the *h +* cell-specific *map2* promoter. Cells were cultured for 24 hr in PM-N+ ade + leu and then plated to SPA+ ade + leu (at 0.5× density) or SPAS (at 1× density) (see figure legends). Plates were incubated at 25°C for 1 or 12 hr (SPA+ ade + leu and SPAS, respectively) before we prepared cells for microscopy using the agar punch method described above. Two microscopes were used for this analysis. In one case, a Nikon Ti microscope coupled to a Yokogawa CSU W1 spinning disc was used. CFP was excited with 445 nm and YFP was excited with 515 nm laser light. Cells were imaged with a 60× Plan Apochromat Objective (NA 1.4). CFP fluorescence was filtered through a 480/30 bandpass filter and YFP was filtered through an ET535/36 m bandpass filter. Fluorescence was recorded with an iXon DU897 EMCCD (Andor). In a separate experiment, a Nikon Ti2-E widefield microscope was used. Here, a bulb was filtered to excite CFP with a 440/20 nm bandpass filter while YFP was excited with a 510/25 nm bandpass filter. The cells were imaged with a 60× Plan Apochromat phase three objective (NA 1.4). The fluorescence of CFP was filtered through an ET472/24 nm bandpass filter and YFP was filtered through a 542/21 nm bandpass filter. Here, images were acquired with a Prime 95 sCMOS camera (Photometrics). In both datasets, images were acquired every 20 min for 24 hr. Images from these datasets were Gaussian blurred one pixel, then rolling ball background subtracted with a radius of 50 pixels. Sample movement was eliminated by registering the images using the plugin 'StackRegJ_' in Fiji. Subsections of the image were created by duplicating out 500 × 500 pixel regions from this large image. These were then Gaussian blurred one pixel again and brightness and contrast adjusted as needed to illustrate the fluorescence. Finally, these small fields of view were cropped again to yield the images in *Figure 2—figure supplement 1*.

## Measuring inbreeding coefficients using genetics

We used a high-throughput system to genotype the progeny from each cross. First, we crossed two parental populations to generate spore progeny as described above. In addition to placing the mixed

haploid cells on SPAS, we also diluted a subset of the mix and plated it onto YEAS (YEA +45 mg/L adenine, histidine, leucine, lysine, and uracil). We genotyped the colonies that grew on that YEAS plate to measure the starting frequency of each parental strain in the cross. We plated the spores produced by the cross on YEAS and grew them at 32°C for 4 days. We picked the colonies using a Qpix 420 Colony Picking System and cultured them in YEL for 24 hr at 30°C in 96-well round-bottom plates (Axygen). We then used a Singer RoTor robot to spot the cultures to YNP dropout and YEAS drug plates and incubated them at 32°C for 3 days. We then imaged the plates using an S&P robotics SPImager with a Canon EOS Rebel T3i camera. We analyzed each picture using the *subtract background* feature in Fiji and assigned regions of interest to the 384 spots where cells were pinned. We then measured the average intensity of each spot and classified cells as grown or not by a heuristic threshold. We genotyped some cross progeny manually using standard techniques due to robot unavailability, with indistinguishable results. A single cross was considered as one biological replicate.

We then inferred the frequency of mating between cells of distinct genotypes based on the frequency of recombinant progeny using a combination of either two or three unlinked genetic markers. If mating was random, we expect the progeny to reflect Hardy-Weinberg expectations $(p^2 + 2p(1-p) + (1-p)^2 = 1)$, where $p^2 + (1-p)^2$ reflect the expected frequency of homozygotes and $2p(1-p)$ reflects the expected frequency of outcrossing. Homozygotes of either parental genotype can only produce offspring with the parental genotypes. If the two parental strains outcross to generate heterozygotes, they will make the parental genotypes and recombinant genotypes all in equal frequencies ($2^n$ total genotypes where $n$ is the number of segregating markers). For our crosses with three markers, we therefore expected the true 'observed' frequency of progeny produced by outcrossing to be equal to the number of observed recombinants divided by 6/8. For our crosses with two unlinked markers, we divided by 2/4. We then calculated the inbreeding coefficient using the formula,

$$F = 1 - \frac{(the\ true\ observed\ fraction\ of\ progeny\ produced\ by\ outcrossing)}{2p(1-p)}.$$

## Zygote frequency expectation under inbreeding and different mating efficiencies

We calculated the expected zygote frequencies when isolates were mated with different isolates on SPA (see Measuring inbreeding coefficients by microscopy) using an additive model that incorporated mating efficiencies and inbreeding coefficients measured from the isogenic crosses. The model assumed that each strain contributes equally to the cross, and that they do not change their own mating in response to the mating partner.

We calculated the expected frequency of homozygotes for parental strain one as:

$$= \left[ p^2 + p(1-p)F_{I1} \right] (1 - u_{s1}),$$

where the inbreeding coefficient is $F_{I1}$ and the mating efficiency is $(1 - u_{s1})$ for parental strain 1 ($s1$), considering its initial frequency $p$. The expected heterozygote frequency is:

$$= 2p(1-p)\left(1 - \frac{F_{I1}+F_{I2}}{2}\right)\left(1 - \frac{u_{s1}+u_{s2}}{2}\right).$$

The expected fraction of homozygotes for parental strain two is:

$$= 1 - (Homozygote\ frequency\ strain\ 1 + Heterozygote\ frequency)$$

## Calculating expected allele frequencies after sexual reproduction

To model the expected changes in allele frequencies in a randomly mating population over time, we used the equations described by *Crow, 1991*. For nonrandomly mating populations, we included the '$F$' inbreeding coefficient in the equations, similar to *Hartl and Clark, 2007*. We assumed that *wtf4* had a drive strength ('$k$') of 0.98 based on experimental observations (*Nuckolls et al., 2017*). We assumed the control allele exhibited Mendelian transmission ($k = 0.5$). Simulations for the spread of a driver in *Figure 3* only considered drive and inbreeding coefficients. To simulate drive in fluorescent populations, the starting frequencies of each allele (i.e., '$p$') were determined empirically for

each experiment using either traditional genetic approaches or cytometry. Predicted values were calculated from each biological replicate (independent cross), composed of three technical replicates measured via cytometry per cross. We assumed all genotypes had equal fitness as haploids. For relative fitness of diploids, we assigned mCherry/mCherry homozygotes a fitness of $w_{11} = 1$, regardless of whether they were EV/EV or *wtf4*/*wtf4* homozygotes. In all but one cross (see below), we observed a fitness cost linked to the GFP alleles relative to mCherry alleles during sexual reproduction, regardless of *wtf4*. We therefore used our data (see below) to calculate 0.234 as the fitness cost of the GFP-linked variant. Because of that, we assigned a fitness value of 0.766 to GFP homozygotes, $w_{22}$. We found the fitness cost linked to GFP had low dominance (see below) and thus assigned a fitness value of 0.98 for GFP/mCherry homozygous for *wtf4* or empty vector, $w_{12}$. For GFP/mCherry heterozygotes that were also heterozygous for *wtf4*, we assigned a fitness of (0.5) that accounts for the death of spores killed by drive (0.51) and the GFP cost (0.234) and dominance (0.083).

The calculation for allele frequency for a *wtf* meiotic driver in consecutive sexual cycles from haploid populations is:

$$p_{g+1} = \frac{\left[ p_g^2 + Fp_g \left( 1-p \right)_g \right] w_{11} + 2p_g \left( 1-p \right)_g k \left( 1-F \right) w_{12}}{\overline{W}_g}.$$

where the mean fitness of the population at each generation (*g*

$$\overline{W}_g = p_g^2 w_{11} + 2p_g \left( 1-p \right)_g w_{12} + \left( 1-p \right)_g^2 w_{22} + Fp_g \left( 1-p \right)_g \left( w_{11} + w_{22} - 2w_{12} \right).$$

When the fitness reduction in heterozygotes is only due to dead cells by drive with no additional costs, $w_{12} = \frac{1}{2k}$, the drivers spread under any initial frequency greater than 0 (*Crow, 1991*).

To determine the fitness cost of the GFP-linked variant that was present in most of our experiments, we used the L-BFGS-B algorithm to find a fitness that maximized the likelihood that our observed allele frequencies varied only due to the linked cost of GFP (*c*) and dominance (*h*) in the first six generations of the control experiments shown in *Figure 4—figure supplement 2* (*Byrd et al., 1995*). To do this, we used the *mle* function from the R *stats4* package (*Team, 2019*). We assumed the fitness cost alters the relative fitness of both homozygotes $w_{22} = 1 - c$ and heterozygotes $w_{12} = 1 - ch$. The fitting was done using only the initial six generations due to a rapid loss of fluorescent cells from seven to ten generations (*Figure 4—figure supplement 1*).

To calculate the minimum initial frequency of driver required to spread in a population when linked to alleles with varying additional fitness costs, we use the additional linked cost (*c* and their dominance (*h*) associated to each genotype and solved using (*Wolfram, 1987*). For simplicity we also assumed complete transmission bias, $k = 1$. A *wtf* meiotic driver linked to a deleterious allele can spread under the condition (critical value) that

$$F_{critical} < \frac{ch(p-1) - cp + p}{c(h-1)(p-1) + p}.$$

where *p* is the initial frequency of a *wtf* driver, *c* is the cost linked to the driver, and *h* is the dominance coefficient. For this case the assigned relative fitness for homozygote carrying the *wtf* driver is $w_{11} = 1 - c$.

## Modeling driver success in finite populations

To determine the possibility of a meiotic driver to spread and be fixed in a finite population, we ran stochastic simulations using a Wright-Fisher model considering multiple population sizes. We assumed no fitness differences between genotypes during haploid cell growth, complete drive (*k* = 1), gamete killing proportional to drive in heterozygotes, no linked deleterious alleles, and non-overlapping generations. We used characters of '0' for non-driving cells and '1' for driving cells (*Figure 3—figure supplement 1B*). Initially the population starts with population size (*N*). We next sampled a fixed population size (*n*) with replacement, assuming infinite gamete supply used for mating (e.g., as if mitotic expansion of the population occurred prior to mating). In our simulations N = n. We sampled mating groups without replacement for an inbreeding population ($S_{pop}$) with a fraction *F* and the rest of cells (n − $S_{pop}$) mate randomly. Populations that mate randomly are proportional to Hardy-Weinberg genotype frequencies with approximated integer values. Homozygotes ('0,0' and '1,1') produce four

progeny ('0,0,0,0' and '1,1,1,1', respectively), while heterozygotes produce two progeny that inherit the drive allele ('1,1'). When a fitness cost is associated with a genotype, we decimated the gametic products in proportion to the cost. The collection of 0's and 1's after this process constituted the next generation. We repeated 1000 generations for 1000 iterations for each tested condition.

## Measuring allele frequencies for experimental evolution analyses

We performed the crosses and collected spores as described above. We then started the next generation by culturing 60 µL of spores from each cross in each of three different wells with 600 µL fresh YEL media in 96 deep-well-round-bottom plates (Axygen) and cultured for 24 hr at 1200 rpm at 32°C. We then transferred 60 µL from each culture to a new plate with 600 µL YEL and cultured for 12–14 hr at 1200 rpm at 32°C. We then pooled the culture replicates in an Eppendorf tube, spun down, and resuspended them in an equal volume of ddH$_2$O. We then took 100 µL of this sample to assay via cytometry (described below). We also plated 200 µL of each sample on SPAS plates and incubated at 25°C for 5 days to allow the cells to mate and sporulate.

To detect and quantify fluorescent cells via flow cytometry, we used the ZE5 Cell Analyzer (Bio-Rad). We spun down 100 µL of each culture, washed the cell pellet with water, spun down again, and resuspended the cells in 200 µL of 1× PBS (phosphate buffered saline) with 1.5 µL 4′,6-diamidino-2-phenylindole (DAPI, Sigma-Aldrich, 100 ng/mL). DAPI stains dead cells, so we considered DAPI-negative cells as live cells. To image DAPI, we used 355 nm laser excitation and a 447/60 nm detector. To image GFP, we excited using a 488 nm laser and detected emission with a 525/35 nm filter. To image mCherry, we excited using a 561 nm laser and detected emission with a 615/24 nm filter. We used 405 and 488 nm (FSC 405 and FSC 488) lasers for forward scatter and 488 nm laser for side scatter (SSC 488).

To quantify the frequency of GFP- or mCherry-positive cells, we analyzed the Flow Cytometry Standard files in R/Bioconductor using the packages FlowTrans and FlowClust (*Lo et al., 2009*). We first separated the cells that had round and uniform shape and similar granularity. This step allowed us to detect a more uniform population of single cells. We then discarded DAPI-positive cells. To determine the GFP- and mCherry-positive cells, we used limits for each channel. The limits varied throughout the experiment due to reconfiguration in the flow cytometer. For every measurement, we corrected with standard samples of cells that only expressed either GFP or mCherry. Fluorescent GFP- and mCherry-positive cells always showed non-overlapping cell populations. We quantified cells that were not classified as non-fluorescent cells.

## Fluorescent marker loss in experimental evolution

In the experimental evolution experiments (*Figures 4 and 5B*, and *Figure 4—figure supplement 2*), a large proportion of cells lost their fluorescent marker over time (*Figure 4—figure supplement 1A-B*). We assume this is because we introduced the markers using an integrating plasmid that enters the genome following a single crossover event. Because of this, a single crossover can then pop the marker and the associated vector out of the genome. We repeatedly observed that the loss occurred faster with the mCherry markers. Consistent with this model, cells that lost fluorescence generally also lost the associated drug resistance marker present on the integrating vector.

We only considered fluorescent cells in our analyses. In the experiments (*Figure 4B–C*, *Figure 4—figure supplement 1A-B*, and *Figure 5B*), we extended the evolution up to 10 generations. In other experiments (*Figure 4D–E* and *Figure 4—figure supplement 2C-D*), we removed non-fluorescent cells by cell sorting after generations 2 and 5. To sort cells, we first collected and cultured spores as described above with the first culture done for 12 hr. We then transferred 60 µL of germinated cells into 600 µL fresh YEL media. We used three cultures for each experimental line and pooled all in equal amounts to have 1.4 mL of each line. We spun each sample down and resuspended the cells in 5 mL ddH$_2$O prior to sorting. We removed non-fluorescent cells and retained GFP-positive (488 nm laser for excitation and a filter 507 nm) and mCherry-positive (561 nm laser for excitation and a 582 nm filter) cells using the laser the BD FACSMelody cell sorter software. We collected 1.2 million cells for each line into 1× PBS. We then spun down the cells and resuspended the pellets in 200 µL YEL. We then took 60 µL from each sample and diluted the cells into 600 µL YEL and continued with the time course as described above. This restored fluorescently labeled cells populations as expected for homothallic and heterothallic lines (*Figure 4—figure supplement 1B*). Cell sorting did not affect our results as we

observed the same patterns in replicate experiments in which we did not remove the non-fluorescent cells (*Figure 4*).

## Software flow cytometry data analysis and modeling scripts

Analysis performed to filter, measure, and compare flow cytometry with simulations are located in the Zanders Lab Github repository, https://github.com/Zanders-Lab/Diverse_mating_phenotypes_impact_the_spread_of_wtf_meiotic_drivers_in_S.-pombe; *Zanders, 2021*, http://www.stowers.org/research/publications/libpb-1625.

## Acknowledgements

We thank members of the Zanders lab, María Bravo Núñez, and Ibrahim M Sabbarini for their helpful comments on the paper. We are grateful to Gerry Smith and Geneviève Thon for sharing various strains. We thank Alexandra Cockrell and Andrew Box for technical support. Original data underlying this manuscript can be accessed from the Stowers Original Data Repository at http://www.stowers.org/research/publications/libpb-1625. This work was performed to fulfill, in part, requirements for JFLH's thesis research in the Graduate School of the Stowers Institute for Medical Research. This work was supported by The Stowers Institute for Medical Research (SEZ); the Searle Scholars Award (SEZ); and the National Institutes of Health (NIH) DP2GM132936 (SEZ). The funders had no role in study design, data collection and analysis, or manuscript preparation. The content is solely the responsibility of the authors and does not necessarily represent the official views of the funders.

## Additional information

### Competing interests

Sarah E Zanders: Inventor on patent application 834 serial 62/491,107 based on wtf killers. The other author declares that no competing interests exist.

### Funding

| Funder | Grant reference number | Author |
|---|---|---|
| Stowers Institute for Medical Research | | Sarah E Zanders |
| National Institute of General Medical Sciences | DP2GM132936 | Sarah E Zanders |
| Searle Scholars Program | | Sarah E Zanders |

The funders had no role in study design, data collection and interpretation, or the decision to submit the work for publication.

### Author contributions

José Fabricio López Hernández, Conceptualization, Data curation, Formal analysis, Investigation, Methodology, Software, Validation, Visualization, Writing – original draft, Writing – review and editing; Rachel M Helston, Conceptualization, Data curation, Formal analysis, Investigation, Methodology, Validation, Visualization, Writing – original draft, Writing – review and editing; Jeffrey J Lange, R Blake Billmyre, Data curation, Formal analysis, Investigation, Methodology, Validation, Visualization, Writing – review and editing; Samantha H Schaffner, Data curation, Formal analysis, Investigation, Writing – review and editing; Michael T Eickbush, Data curation, Formal analysis, Investigation, Methodology, Writing – review and editing; Scott McCroskey, Formal analysis, Investigation, Methodology, Writing – review and editing; Sarah E Zanders, Conceptualization, Formal analysis, Funding acquisition, Methodology, Project administration, Supervision, Visualization, Writing – review and editing

### Author ORCIDs

José Fabricio López Hernández 🔗 http://orcid.org/0000-0003-2633-3690
Rachel M Helston 🔗 http://orcid.org/0000-0001-6022-6755
R Blake Billmyre 🔗 http://orcid.org/0000-0003-4866-3711

Michael T Eickbush http://orcid.org/0000-0001-9057-9156
Sarah E Zanders http://orcid.org/0000-0003-1867-986X

**Decision letter and Author response**
Decision letter https://doi.org/10.7554/eLife.70812.sa1
Author response https://doi.org/10.7554/eLife.70812.sa2

## Additional files

### Supplementary files
• Supplementary file 1. Yeast strains used in this study. Strains used and created in this study are listed.
• Supplementary file 2. Plasmids used in this study.
• Supplementary file 3. Oligo table. Oligos used in this study.
• Transparent reporting form

### Data availability
Original data underlying this manuscript can be accessed from the Stowers Original Data Repository at http://www.stowers.org/research/publications/libpbxxxx. Base called reads are available as fastq files at the SRA under project accession number PRJNA732453.

The following dataset was generated:

| Author(s) | Year | Dataset title | Dataset URL | Database and Identifier |
|-----------|------|---------------|-------------|-------------------------|
| Zanders SE | 2021 | NCBI BioProject | https://www.ncbi.nlm.nih.gov/bioproject/?term=PRJNA732453 | NCBI BioProject, PRJNA732453 |

The following previously published datasets were used:

| Author(s) | Year | Dataset title | Dataset URL | Database and Identifier |
|-----------|------|---------------|-------------|-------------------------|
| Eickbush MT, Young JM, Zanders SE | 2018 | Mate-pair sequencing of S. kambucha lab strain SZY661 | https://www.ncbi.nlm.nih.gov/sra/SRX4224586 | NCBI Sequence Read Archive, SRX4224586 |

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
