## [Editor Report]

Meiotic drivers are selfish elements that distort segregation to be over-represented in offspring of heterozygotes. Multiple meiotic drive elements are known in the yeast *Schizosaccharomyces pombe, S. pombe*, which can seem puzzling as this fungus has long been thought to undergo little outcrossing. This manuscript reports theoretical and experimental analyses suggesting that the outcrossing rate can be high enough in this species to explain the spread of multiple meiotic drive elements. The findings support the emerging view that homothallic fungi can undergo quite high rates of outcrossing, which is also in agreement with evolutionary considerations on the evolution of mating types. This study can thus be of high relevance for scientists studying meiotic drivers and/or mating systems and their evolution.

---

## [Decision Letter]

**Decision letter after peer review:**

Thank you for submitting your article "Diverse mating phenotypes impact the spread of wtf meiotic drivers in *S. pombe*" for consideration by *eLife*. Your article has been reviewed by 3 peer reviewers, including Tatiana Giraud as the Reviewing Editor and Reviewer #1, and the evaluation has been overseen by Patricia Wittkopp as the Senior Editor. The following individual involved in review of your submission has agreed to reveal their identity: Bart Nieuwenhuis (Reviewer #3).

Essential revisions:

As detailed in the reviews below, the major essential revisions are the following ones, requiring further modeling and experiments :

1) The introduction and discussion should be extensively revised to better use and define terms such as selfing, inbreeding and outcrossing and better place the study in its evolutionary context

2) Population genetic modeling should be used to make more ambitious predictions about the natural history of meiotic drive in *S. pombe* and other organisms.

3) The claim that mating-type switching frequencies affects outcrossing is not substantiated and the model cannot explain how drive overcomes drift after the rise of a novel allele. This should be formally modeled in a structural environment and/or tested experimentally

4) The inferences of the relative fitness cost of GFP should be toned down.

The referees also provide a list of excellent additional suggestions, which should be addressed as well if you choose to resubmit the manuscript.

*Reviewer #1 (Recommendations for the authors):*

The whole introduction is based on the rationale that homothallism means selfing while heterothallism means outcrossing. There are several issues with such assumptions. First selfing and outcrossing will be understood by most readers of e*Life* as “diploid selfing/outcrossing”, as in plants and animals, while heterothallism does not prevent diploid selfing, and actually many heterothallic fungi undergo mostly diploid selfing (e.g. Microbotryum violacem, Neurospora tetrasperma, Podospora anserina). Conversely, homothallism does not prevent outcrossing and may not even have evolved to facilitated inbreeding but instead the opposite, universal compatibility under outcrossing. See below some references discussing these issues and some references showing that homothallic fungi do outcross and that many heterothallic fungi do self a lot. I would therefore not consider particularly surprising that *S. pombe* outcross enough for meiotic drivers to spread, although this indeed goes against the current view, based on the misconceptions above. The only good way to estimate outcrossing rates is by using population genetics or genomics on natural populations, as experiments will likely fail to mimic natural dispersal or conditions of mating. This is briefly discussed in the discussion but the whole introduction is misleading in my opinion. I understand this is a quite frequent view among scientists working on fungi (as they infer rates based on what they observe in Petri dishes and were often not evolutionary biologists), but the authors precisely miss the opportunity to state that this view is likely incorrect and to explain why, as this is precisely what they show. I did not understand why outcrossing in natural populations in *S. pombe* has not been estimated or at least discussed based on the genomic data in Jeffares et al. 2015. I think that framing the study into this evolutionary context will only make it stronger. I would also recommend to highlight much more the variation in the propensity of inbreeding among natural isolates and the existence of heterothallic strains. Is it frequently the case in nature?

doi: 10.1111/j.1420-9101.2012.02495.x.

10.1111/j.1469-185X.2010.00153.x.

10.1128/EC.00440-07

10.1038/hdy.2014.37

10.1111/nph.17039

The “selfing” in the manuscript would be better be named as “same-clone mating”, whose evolutionary consequences are drastically different from more classical diploid selfing (10.1111/j.1420-9101.2012.02495.x).

*Reviewer #2 (Recommendations for the authors):*

As it stands, the population genetic modeling (Main Text lines 315-339, Methods lines 727-773, Figures3,5A) is used mainly to generate frequency trajectories of meiotic drivers against which the trajectories observed in the evolution experiments (Figures4,5B) can be compared. Of course, the similarity of the predicted and experimentally observed trajectories is exciting – I suspect that, based on results like these, pombe will soon become (if it is not already) the premier experimental system for meiotic drive. Still, it seems a missed opportunity, given the variation in mating behavior across strains that the authors have so carefully documented, not to use the population genetic modeling to link this variation with (perhaps even immediately testable) predictions about the natural history of drive in pombe. Below, I suggest some directions in which this might be achieved.

– The authors focus on the effect of inbreeding on the spread of a driver that starts at an appreciable frequency in the population (e.g., 5%, as in Figure 3B). They find that, while inbreeding slows down the spread of a driver from such frequencies, it does not prevent the driver's eventual spread to fixation. Based on this result, a reader might naively infer that inbreeding affects only the transient dynamics of meiotic drive, not whether a driver spreads to fixation or not; and therefore that strains with different levels of inbreeding will not necessarily have different numbers of fixed drivers.

However, a new driver appears by mutation as a single copy at its locus, and from this starting point, the driver's spread to fixation is not guaranteed – indeed, in the case of spore killers in fungi, it is always unlikely (Martinossi-Allibert et al. 2021). The probability that the driver will go on to achieve an appreciable frequency in the population (and therefore eventually to fix) is governed by the interplay between random stochastic drift at low copy numbers of the driver and the deterministic advantage enjoyed by the driver. Since the latter depends strongly on the degree of inbreeding, inbreeding will substantially affect the fixation probability, and therefore the substitution rate, of meiotic drivers in the various strains studied by the authors. For example, if outcrossing occurs once every 100,000 generations, then a driver starting at low copy number is subject to ~100,000 generations of random drift before it can benefit from its transmission advantage; in this long interval, the driver is very likely to have drifted to extinction. In contrast, in a strain that outcrosses every generation, a newly arisen driver exercises its transmission advantage immediately and regularly, and therefore has a much higher chance of escaping stochastic loss.

I believe that questions like these could be investigated using the authors' population genetic model simply by simulating stochastic versions of this model – say, under a Wright-Fisher process – and starting the simulations with a single copy of the driver. This exercise would, I suspect, yield potentially testable predictions about variation in the number of fixed wtf drivers across different strains. Of course, these predictions would be "all else equal", but it would nevertheless be interesting just to see – given that strains do vary in the number of fixed wtf drivers (lines 490-491) – whether strains with lower levels of inbreeding harbor more fixed drivers.

– Turning to the transient dynamics of drivers on their way to fixation, it is striking in Figure 3B just how severely inbreeding affects the time required for fixation of a driver. It seems that the trajectories should be "stretched" according to the number of generations until each outcrossing event. Thus, in a strain that outcrosses once every 100,000 generations, a driver will take ~100,000x more generations to fix than it would in a strain that outcrosses every generation. This yields the prediction that, in strains with higher levels of inbreeding, although there should be fewer drivers fixed or spreading towards fixation (above), a greater fraction of these drivers should still be on their way to fixation, relative to a strain with lower levels of inbreeding. I.e., a greater fraction of drivers should still be polymorphic in strains that inbreed more frequently. (This prediction depends, I think, on the rate of appearance of drivers in the population relative to the average time a driver takes to fix.)

Empirically, are polymorphisms ever observed for wtf drivers? If not, why not? If so, do strains differ in the fraction of driver loci that are polymorphic vs fixed? If the answers to these questions are interesting, the authors might wish to use stochastic simulations of their population genetic model to obtain a more precise understanding of the predicted differences in polymorphism across strains.

– Finally, it has recently been suggested that, owing to inbreeding's effect in slowing the spread of meiotic drivers, the presence of meiotic drivers in a species might conversely select for greater degrees of inbreeding (Bull 2017). This selective force would appear to be especially strong in pombe, in which drivers are both pervasive and highly destructive. Perhaps the authors could discuss whether they believe that mating behavior across strains of pombe (and perhaps further afield) have been shaped by meiotic drivers?

References:

Bull JJ. (2017). Lethal gene drive selects inbreeding. Evolution, Medicine, and Public Health, 2017(1), 1-16.

Drury DW, Dapper AL, Siniard DJ, Zentner GE, and Wade MJ. (2017). CRISPR/Cas9 gene drives in genetically variable and nonrandomly mating wild populations. Science Advances, 3(5), e1601910.

Hamilton WD. (1967). Extraordinary sex ratios. Science, 156(3774), 477-488.

Martinossi‐Allibert I, Veller C, Ament‐Velásquez SL, Vogan AA, Rueffler C, and Johannesson H. (2021). Invasion and maintenance of meiotic drivers in populations of ascomycete fungi. Evolution, 75(5), 1150-1169.

Nauta MJ and Hoekstra RF. (1993). Evolutionary dynamics of spore killers. Genetics, 135(3), 923-930.

*Reviewer #3 (Recommendations for the authors):*

I have enjoyed the paper and am impressed with the quality and amount of work performed.

A few recommendations:

1. Measure the mating type switching frequency directly using fluorescent markers.

2. Rework the manuscript. It currently reads very chronological there are at least 5 paragraphs that start with 'we next'. Write more from the question, and less from the experiment.

3. Indicate the total number of strains for which genetic manipulations were tried.

4. The model needs to be written out a bit better. For example, it is not clear without reading Crow 1991 what p stands for or how F is defined here.

5. It would be interesting to see what happens if you add a small direct fitness benefit to the driver allele, to the model, but especially experimentally!

6. The number of replicates is not clearly indicated for all experiments.

7. The recent fungal spore-killer literature outside of fission yeast needs to be incorporated into this paper, especially the recent theoretical work by Martinossi-Allibert et al. 2021.

[Editors' note: further revisions were suggested prior to acceptance, as described below.]

Thank you for resubmitting your work entitled "Diverse mating phenotypes impact the spread of wtf meiotic drivers in *S. pombe*" for further consideration by *eLife*. Your revised article has been reviewed by 3 peer reviewers, one of whom is a member of our Board of Reviewing Editors, and the evaluation has been overseen by Patricia Wittkopp as the Senior Editor.

The referees found the manuscript improved, but had still serious concerns, in particular about the use of terms such as outcrossing, inbreeding and fitness costs. It is very important to use unambiguous and exact terms in science. Referee 3 further suggests some clarification, discussion and possibly simulations on fitness costs or benefits of the driving allele and on the effect of drift.

*Reviewer #1 (Recommendations for the authors):*

I found the manuscript much improved, and the additional analyses interesting.

However, I think that the use of “inbreeding” and “outcrossing” are still unclear and misleading. Many readers will just read this as typical inbreeding and outcrossing in plants and animals while the phenomena here are completely different. In addition, you sometimes use “outcrossing” as true outcrossing (eg L94, L114) and at many places one wonders in what sense you use the terms (in particular iin the model part). Science should use exact terms and avoid confusion and ambiguity. I think that “inbreeding” should be replaced by “same-clone mating” and “outcrossing” by something like “non-same-clone mating”. In addition, as said in my previous review, more emphasis should be put on the high rate of non-same-clone mating in homothallics, and how this goes against the (wrong) view in the fungal literature ; this is what is important for the spread of drivers. The emphasis is currently put on the lower rate of non-same-clone mating in heterothallics which is just its definition. These question should be specifically written in the introduction, refering to previous publication showing that homothallics do outcross (e.g., 10.1038/hdy.2014.37). The discussion should also more explicitly discuss the fact that, in natural conditions, dispersal may promote further non-same-clone mating and (true) outcrossing.

In addition, I found the answer to the previous comment below not convincing, this should be more explicitly discussed : “This method introduces a bias towards the correlation switching and selfing, because the latter is used as a proxy for the first.”

*Reviewer #2 (Recommendations for the authors):*

The authors have addressed my previous comments, in some cases carrying out interesting new analyses that have improved the manuscript. I appreciate that some of the questions raised in my previous report concerning the long-term population genetics of drive in pombe are beyond the scope of the present work, though I encourage the authors to consider these questions in future work since they are, I think, "within reach" and would lead to greater understanding of the fantastic system this group is developing.

*Reviewer #3 (Recommendations for the authors):*

The revision of the authors has greatly improved the manuscript and has removed most of the questions I had placed with the previous version. The logic of the interaction between switching frequency, outcrossing and population dynamics are better explained and caveats to the conclusions are addressed. The addition of the information on the multiallelic wtf genes also helps with understanding how allele frequencies change over time and that an established allele does not mean fixation. Below a few additional comments on the new manuscript and some elaborations on my previous comments.

The authors have chosen to consistently use the term selfing to describe same clone mating. Even though they clearly define the use of their term, I disagree with this choice. To a population geneticist or evolutionary biologist this is confusing. The term 'same clone mating' is unambiguous and would be preferred.

It is unfortunate that direct measurements of mating types was not possible. Though sad that this effort was unsuccessful, it is personally comforting to hear, as in our hands tracking of switching using fluorescent markers has also been unsuccessful. It would be great if a measure would be possible and hopefully a good method will become available to directly assess switching.

From the reply to my comments 5 and 6 it is clear I did not express myself clearly. Specifically, the use of the fitness costs or benefits of the driving allele are not interpreted as I meant them. Let me elaborate on this. There are different moments where a cost can be incurred during the lifecycle of fission yeast. First during the haploid phase (e.g. germination, haploid mitosis), during mating and at the diploid phase (e.g. diploid mitosis, meiosis, sporogenesis). The fitness in the haploid phase (not modelled by the authors) is likely based on the genome of the individual, whereas mating and the diploid phase are affected by the two genomes. When I refer to the fitness cost/benefit, I always refer to the effect on an allele, as this is the unit of inheritance (similar to the change in p modelled by the authors). The 'cost of a driver' would be the negative effect the allele has on its reproduction, either during the haploid or diploid phase (changing w22 and/or w12 in the model). Killing per se is not a cost to the driver, as it does not reduce its own survival when heterozygous, it only affects the wild type allele. There is a cost to the diploid individual (fitness = 0.5), causing genomic conflict, but under full drive (k=1) this does not affect the driving allele and is thus not a cost to this allele. Drivers can become associated with deleterious mutations due to suppressed recombination, gene disruptions etc. as the authors also mention in line 599, however, to my knowledge such genetic associations have not been described in fission yeast. It might be good to explicitly mention the situation for fission yeast.

Also no reduction in haploid or diploid growth rate has been shown for drivers, or at least not for wtf4 ("Outside of its role in meiotic drive, wtf4 has no apparent role in promoting fertility", pp. 13 Nuckolis et al. 2017) and therefore I consider these driving alleles neutral.

I asked before if there could be a direct benefit to driving alleles. The most obvious would be a benefit to germination rate/ spore survival due to potential resource allocation from the killed spores to the killers. This could improve fitness during the haploid phase (w1 > w2), thereby increasing the allele frequency of the drivers before the next sexual phase. Such secondary effects of killing could greatly speed up the initial phase of invasion when the driver's frequency is low and killing thus slow, which helps to overcome the selection-drift barrier.

Staying with drift. It is nice to see the extension of the model using different population sizes (lines 500-512; Figure S3c). The results that show that drivers have a higher chance of establishment in smaller populations fit with prediction from the Wright-Fisher model where chance of fixation is directly proportional to the allele frequency. With smaller populations, the initial allele frequency increases (1/N) reducing the chance of loss by drift. It is not clear to me how the "local patch" dynamics as discussed in lines 741-746 affects the dynamics at the population level, where the allele frequencies of all the local patches are combined resulting in the initial population with large population sizes. If there is some form of population structure, of which there is probably some (Jeffares et al. 2015, Tusso et al. 2019) the allele frequencies might locally become higher increasing the chance of establishment of novel alleles. It would be interesting to see how multiple small demes with migration might affect speed of invasion and might affect dynamics of multiple drivers.

---

## [Author Response]

Essential revisions:As detailed in the reviews below, the major essential revisions are the following ones, requiring further modeling and experiments :1) The introduction and discussion should be extensively revised to better use and define terms such as selfing, inbreeding and outcrossing and better place the study in its evolutionary context

We have made these revisions.

2) Population genetic modeling should be used to make more ambitious predictions about the natural history of meiotic drive in *S. pombe* and other organisms.

We have extended the population genetic modeling.

3) The claim that mating-type switching frequencies affects outcrossing is not substantiated and the model cannot explain how drive overcomes drift after the rise of a novel allele. This should be formally modeled in a structural environment and/or tested experimentally

Under the conditions used in our study, mating type switching does affect outcrossing in

*Sp*. Homothallic *Sp* strains inbreed whereas our control mixed heterothallic *Sp* populations mate more randomly (Figure 1D, Figure 1—figure supplement 2). Inbreeding of homothallic cells is something *Sp* geneticists have long assumed to be true and we formally quantify. We do not claim that homothallism in fungi *per se* leads to inbreeding. The new manuscript makes that more clear.

The claim that *Sk* has a lower switching rate was first made by Singh and Klar, 2002.

These authors claimed there was lower switching based on the observation of less switching-inducing DSB. It is hard to envision how switching rates could not fall with reduced DSB, but we still performed additional analyses using experiments akin to those originally used to characterize switching rates in *Sp*. All of our data are consistent with decreased switching in *Sk*. Still, we explicitly present decreased switching in *Sk* as a model that is consistent with the data, but is not proven.

Finally, our new population genetic analyses incorporate drift and our revised discussion also further discusses conditions under which novel *wtf* drivers can become established.

4) The inferences of the relative fitness cost of GFP should be toned down.

We toned down the discussion of the GFP fitness costs and used a better approach to

estimate those costs and their dominance.

Reviewer #1 (Recommendations for the authors):The whole introduction is based on the rationale that homothallism means selfing while heterothallism means outcrossing.

Thank you for pointing this out, as this was not our intent and we are aware that the statement is inaccurate. We have rewritten the introduction to better illustrate our intended narrative that it is believed that homothallic *S. pombe* preferentially undergoes haploid selfing and how this notion leads to considerable confusion about how the species can harbor so many meiotic drivers.

There are several issues with such assumptions. First selfing and outcrossing will be understood by most readers of eLife as “diploid selfing/outcrossing”, as in plants and animals, while heterothallism does not prevent diploid selfing, and actually many heterothallic fungi undergo mostly diploid selfing (e.g. Microbotryum violacem, Neurospora tetrasperma, Podospora anserina). Conversely, homothallism does not prevent outcrossing and may not even have evolved to facilitated inbreeding but instead the opposite, universal compatibility under outcrossing. See below some references discussing these issues and some references showing that homothallic fungi do outcross and that many heterothallic fungi do self a lot. I would therefore not consider particularly surprising that *S. pombe* outcross enough for meiotic drivers to spread, although this indeed goes against the current view, based on the misconceptions above.

Our revised introduction makes clearer that our focus is specifically on phenotypes of hetero- and homothallism in *S. pombe.* We also include references to support the fact that homothallism does not prevent outcrossing in *S. pombe* or other fungal species. Finally, we provide a new figure to illustrate sexual reproduction in *S. pombe* (Figure 1—figure supplement 1).

The only good way to estimate outcrossing rates is by using population genetics or genomics on natural populations, as experiments will likely fail to mimic natural dispersal or conditions of mating.

Please see response to reviewer point 1 in the public review.

This is briefly discussed in the discussion but the whole introduction is misleading in my opinion. I understand this is a quite frequent view among scientists working on fungi (as they infer rates based on what they observe in Petri dishes and were often not evolutionary biologists), but the authors precisely miss the opportunity to state that this view is likely incorrect and to explain why, as this is precisely what they show. I did not understand why outcrossing in natural populations in *S. pombe* has not been estimated or at least discussed based on the genomic data in Jeffares et al. 2015. I think that framing the study into this evolutionary context will only make it stronger. I would also recommend to highlight much more the variation in the propensity of inbreeding among natural isolates and the existence of heterothallic strains. Is it frequently the case in nature?doi: 10.1111/j.1420-9101.2012.02495.x.10.1111/j.1469-185X.2010.00153.x.10.1128/EC.00440-0710.1038/hdy.2014.3710.1111/nph.17039

We have revised the introduction and discussion to address the use of population genetic analyses to predict outcrossing frequencies. We have better highlighted the existence of heterothallic *S. pombe* strains in the introduction. Finally, we have extended our discussion of *wtf* gene evolution.

The “selfing” in the manuscript would be better be named as “same-clone mating”, whose evolutionary consequences are drastically different from more classical diploid selfing (10.1111/j.1420-9101.2012.02495.x).

We do not use the term selfing. In our revision, we provide a more explicit explanation of *S. pombe* sexual reproduction and the terms we use in the introduction and provide a new figure to help illustrate the text (Figure 1—figure supplement 1).

Reviewer #2 (Recommendations for the authors):As it stands, the population genetic modeling (Main Text lines 315-339, Methods lines 727-773, Figures 3, 5A) is used mainly to generate frequency trajectories of meiotic drivers against which the trajectories observed in the evolution experiments (Figures4,5B) can be compared. Of course, the similarity of the predicted and experimentally observed trajectories is exciting – I suspect that, based on results like these, pombe will soon become (if it is not already) the premier experimental system for meiotic drive. Still, it seems a missed opportunity, given the variation in mating behavior across strains that the authors have so carefully documented, not to use the population genetic modeling to link this variation with (perhaps even immediately testable) predictions about the natural history of drive in pombe. Below, I suggest some directions in which this might be achieved.– The authors focus on the effect of inbreeding on the spread of a driver that starts at an appreciable frequency in the population (e.g., 5%, as in Figure 3B). They find that, while inbreeding slows down the spread of a driver from such frequencies, it does not prevent the driver's eventual spread to fixation. Based on this result, a reader might naively infer that inbreeding affects only the transient dynamics of meiotic drive, not whether a driver spreads to fixation or not; and therefore that strains with different levels of inbreeding will not necessarily have different numbers of fixed drivers.However, a new driver appears by mutation as a single copy at its locus, and from this starting point, the driver's spread to fixation is not guaranteed – indeed, in the case of spore killers in fungi, it is always unlikely (Martinossi-Allibert et al. 2021). The probability that the driver will go on to achieve an appreciable frequency in the population (and therefore eventually to fix) is governed by the interplay between random stochastic drift at low copy numbers of the driver and the deterministic advantage enjoyed by the driver. Since the latter depends strongly on the degree of inbreeding, inbreeding will substantially affect the fixation probability, and therefore the substitution rate, of meiotic drivers in the various strains studied by the authors. For example, if outcrossing occurs once every 100,000 generations, then a driver starting at low copy number is subject to ~100,000 generations of random drift before it can benefit from its transmission advantage; in this long interval, the driver is very likely to have drifted to extinction. In contrast, in a strain that outcrosses every generation, a newly arisen driver exercises its transmission advantage immediately and regularly, and therefore has a much higher chance of escaping stochastic loss.I believe that questions like these could be investigated using the authors' population genetic model simply by simulating stochastic versions of this model – say, under a Wright-Fisher process – and starting the simulations with a single copy of the driver. This exercise would, I suspect, yield potentially testable predictions about variation in the number of fixed wtf drivers across different strains. Of course, these predictions would be "all else equal", but it would nevertheless be interesting just to see – given that strains do vary in the number of fixed wtf drivers (lines 490-491) – whether strains with lower levels of inbreeding harbor more fixed drivers.

Thank you for this suggestion as it has expanded our research in exciting ways. We have now added simulation analyses to the paper that consider driver invasion in populations of various population sizes under varying degrees of outcrossing. We hope to expand these analyses in future work to address scenarios that consider: multiple meiotic drivers, suppressors, varying levels of linkage between the drivers/suppressors and compares the fate of drivers on chromosomes that tolerate to that of drivers on chromosomes that do not tolerate aneuploidy. These analyses are ongoing and we believe best suited for a separate study.

– Turning to the transient dynamics of drivers on their way to fixation, it is striking in Figure 3B just how severely inbreeding affects the time required for fixation of a driver. It seems that the trajectories should be "stretched" according to the number of generations until each outcrossing event. Thus, in a strain that outcrosses once every 100,000 generations, a driver will take ~100,000x more generations to fix than it would in a strain that outcrosses every generation. This yields the prediction that, in strains with higher levels of inbreeding, although there should be fewer drivers fixed or spreading towards fixation (above), a greater fraction of these drivers should still be on their way to fixation, relative to a strain with lower levels of inbreeding. I.e., a greater fraction of drivers should still be polymorphic in strains that inbreed more frequently. (This prediction depends, I think, on the rate of appearance of drivers in the population relative to the average time a driver takes to fix.)Empirically, are polymorphisms ever observed for wtf drivers? If not, why not? If so, do strains differ in the fraction of driver loci that are polymorphic vs fixed? If the answers to these questions are interesting, the authors might wish to use stochastic simulations of their population genetic model to obtain a more precise understanding of the predicted differences in polymorphism across strains.

This is an important, but complicated point that we omitted from the original submission that we now include in the discussion. There is one locus (*wtf4*) where all strains that have been surveyed contain a *wtf* meiotic driver, so some may consider that fixed. However, each of those strains contains a distinct allele of that gene, so they function as unique drivers. For example, *Sk wtf4* provides spores no protection against killing of *Sp wtf4* and vice versa. Because of that, we do not consider *wtf4* or any other *wtf* driver fixed in *S. pombe*. Instead, the mutation rate of *wtf* genes into functionally novel *wtf* genes seems to outpace the fixation rate of a driver.

– Finally, it has recently been suggested that, owing to inbreeding's effect in slowing the spread of meiotic drivers, the presence of meiotic drivers in a species might conversely select for greater degrees of inbreeding (Bull 2017). This selective force would appear to be especially strong in pombe, in which drivers are both pervasive and highly destructive. Perhaps the authors could discuss whether they believe that mating behavior across strains of pombe (and perhaps further afield) have been shaped by meiotic drivers?

This is a good point and a good reference. Thank you for bringing it to our attention. We

have added discussion these points to the paper.

[Editors' note: further revisions were suggested prior to acceptance, as described below.]

The referees found the manuscript improved, but had still serious concerns, in particular about the use of terms such as outcrossing, inbreeding and fitness costs. It is very important to use unambiguous and exact terms in science. Referee 3 further suggests some clarification, discussion and possibly simulations on fitness costs or benefits of the driving allele and on the effect of drift.Reviewer #1 (Recommendations for the authors):I found the manuscript much improved, and the additional analyses interesting.However, I think that the use of “inbreeding” and “outcrossing” are still unclear and misleading. Many readers will just read this as typical inbreeding and outcrossing in plants and animals while the phenomena here are completely different. In addition, you sometimes use “outcrossing” as true outcrossing (eg L94, L114) and at many places one wonders in what sense you use the terms (in particular iin the model part). Science should use exact terms and avoid confusion and ambiguity. I think that “inbreeding” should be replaced by “same-clone mating” and “outcrossing” by something like “non-same-clone mating”.

We reviewed each occurrence of these terms and used ‘same-clone mating’ and ‘non-same-clone mating’ where appropriate.

In addition, more emphasis should be put on the high rate of non-same-clone mating in homothallics, and how this goes against the (wrong) view in the fungal literature ; this is what is important for the spread of drivers. The emphasis is currently put on the lower rate of non-same-clone mating in heterothallics which is just its definition. These question should be specifically written in the introduction, refering to previous publication showing that homothallics do outcross (e.g., 10.1038/hdy.2014.37).

We have added to the introduction and discussion to further emphasize that outcrossing is possible in homothallics and to cite papers discussing and demonstrating homothallic outcrossing.

The discussion should also more explicitly discuss the fact that, in natural conditions, dispersal may promote further non-same-clone mating and (true) outcrossing.

We added to the discussion to introduce spore dispersal in *S. pombe* and how that could contribute to outcrossing.

In addition, I found the answer to the previous comment below not convincing, this should be more explicitly discussed : “This method introduces a bias towards the correlation switching and selfing, because the latter is used as a proxy for the first.”

We added to the discussion to introduce spore dispersal in *S. pombe* and how that could contribute to outcrossing.

Reviewer #3 (Recommendations for the authors):The revision of the authors has greatly improved the manuscript and has removed most of the questions I had placed with the previous version. The logic of the interaction between switching frequency, outcrossing and population dynamics are better explained and caveats to the conclusions are addressed. The addition of the information on the multiallelic wtf genes also helps with understanding how allele frequencies change over time and that an established allele does not mean fixation. Below a few additional comments on the new manuscript and some elaborations on my previous comments.The authors have chosen to consistently use the term selfing to describe same clone mating. Even though they clearly define the use of their term, I disagree with this choice. To a population geneticist or evolutionary biologist this is confusing. The term 'same clone mating' is unambiguous and would be preferred.

We have adopted the preferred terms.

It is unfortunate that direct measurements of mating types was not possible. Though sad that this effort was unsuccessful, it is personally comforting to hear, as in our hands tracking of switching using fluorescent markers has also been unsuccessful. It would be great if a measure would be possible and hopefully a good method will become available to directly assess switching.

Agreed!

From the reply to my comments 5 and 6 it is clear I did not express myself clearly. Specifically, the use of the fitness costs or benefits of the driving allele are not interpreted as I meant them. Let me elaborate on this. There are different moments where a cost can be incurred during the lifecycle of fission yeast. First during the haploid phase (e.g. germination, haploid mitosis), during mating and at the diploid phase (e.g. diploid mitosis, meiosis, sporogenesis). The fitness in the haploid phase (not modelled by the authors) is likely based on the genome of the individual, whereas mating and the diploid phase are affected by the two genomes. When I refer to the fitness cost/benefit, I always refer to the effect on an allele, as this is the unit of inheritance (similar to the change in p modelled by the authors). The 'cost of a driver' would be the negative effect the allele has on its reproduction, either during the haploid or diploid phase (changing w22 and/or w12 in the model). Killing per se is not a cost to the driver, as it does not reduce its own survival when heterozygous, it only affects the wild type allele. There is a cost to the diploid individual (fitness = 0.5), causing genomic conflict, but under full drive (k=1) this does not affect the driving allele and is thus not a cost to this allele. Drivers can become associated with deleterious mutations due to suppressed recombination, gene disruptions etc. as the authors also mention in line 599, however, to my knowledge such genetic associations have not been described in fission yeast. It might be good to explicitly mention the situation for fission yeast.Also no reduction in haploid or diploid growth rate has been shown for drivers, or at least not for wtf4 ("Outside of its role in meiotic drive, wtf4 has no apparent role in promoting fertility", pp. 13 Nuckolis et al. 2017) and therefore I consider these driving alleles neutral.

Thank you for these clarifications. We have clarified in the new draft that drive benefits the *wtf* drive allele, but is costly to the fitness of the genome as a whole because meiotic products bearing alleles not linked in cis to the driver are destroyed ~half the time by drive in heterozygotes (e.g. Crow 1991). We specify fitness that our fitness values apply to the diploid stage. We have also specifically addressed what is known about linkage of *S. pombe* drivers to deleterious alleles.

I asked before if there could be a direct benefit to driving alleles. The most obvious would be a benefit to germination rate/ spore survival due to potential resource allocation from the killed spores to the killers. This could improve fitness during the haploid phase (w1 > w2), thereby increasing the allele frequency of the drivers before the next sexual phase. Such secondary effects of killing could greatly speed up the initial phase of invasion when the driver's frequency is low and killing thus slow, which helps to overcome the selection-drift barrier.

We added a Figure 5-Supplementary figure 2 where we modeled a higher fitness of driver heterozygotes such that they gain additional fitness benefits beyond that enjoyed by killing the competing allele. The modeling under this assumption did not fit our data as well as assuming the drivers benefit is exclusively from spore killing.

Staying with drift. It is nice to see the extension of the model using different population sizes (lines 500-512; Figure S3c). The results that show that drivers have a higher chance of establishment in smaller populations fit with prediction from the Wright-Fisher model where chance of fixation is directly proportional to the allele frequency. With smaller populations, the initial allele frequency increases (1/N) reducing the chance of loss by drift. It is not clear to me how the "local patch" dynamics as discussed in lines 741-746 affects the dynamics at the population level, where the allele frequencies of all the local patches are combined resulting in the initial population with large population sizes. If there is some form of population structure, of which there is probably some (Jeffares et al. 2015, Tusso et al. 2019) the allele frequencies might locally become higher increasing the chance of establishment of novel alleles. It would be interesting to see how multiple small demes with migration might affect speed of invasion and might affect dynamics of multiple drivers.

Our intent was to highlight the same point you make about population structure allowing local allele frequencies to become higher and increase the probability of establishment. We reworded this section to make that clearer and also to highlight that Martinossi-Allibert et al. formally described this idea. We agree that testing this idea further is important and hope to pursue that experimentally in future studies.